
# The flood of June 2013 in Germany: how much do we know about its impacts?

A. H. Thieken[1], T. Bessel[2], S. Kienzler[1], H. Kreibich[3,6], M. Müller[4], S. Pisi[5], and K. Schröter[3,6]

[1]University of Potsdam, Institute of Earth and Environmental Science, Karl-Liebknecht-Strasse 24-25, 14476 Potsdam, Germany
[2]Karlsruhe Institute of Technology, Institute for Economics (ECON), Waldhornstrasse 27, 76131 Karlsruhe, Germany
[3]Helmholtz Centre Potsdam, GFZ German Research Centre for Geosciences, Section 5.4 Hydrology, Telegrafenberg, 14473 Potsdam, Germany
[4]Deutsche Rückversicherung AG, NatCat-Center, Hansaallee 177, 40549 Düsseldorf, Germany
[5]German Committee for Disaster Reduction (DKKV), Friedrich-Ebert-Allee 38, 53113 Bonn, Germany
[6]CEDIM – Center for Disaster Management and Risk Reduction Technology, Germany

**NHESSD**

doi:10.5194/nhess-2015-324

The flood of June 2013 in Germany: how much do we know about its impacts?

A. H. Thieken et al.

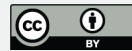

Received: 25 November 2015 – Accepted: 2 December 2015 – Published: 15 January 2016

Correspondence to: A. H. Thieken (thieken@uni-potsdam.de)

Published by Copernicus Publications on behalf of the European Geosciences Union.

**NHESSD**

doi:10.5194/nhess-2015-324

**The flood of June 2013 in Germany: how much do we know about its impacts?**

A. H. Thieken et al.

Interactive Discussion

Discussion Paper | Discussion Paper | Discussion Paper | Discussion Paper

**NHESSD**

doi:10.5194/nhess-2015-324

**The flood of June 2013 in Germany: how much do we know about its impacts?**

A. H. Thieken et al.

## Abstract

In June 2013, widespread flooding and consequent damage and losses occurred in Central Europe, especially in Germany. The paper explores what data is available to investigate the adverse impacts of the event, what kind of information can be retrieved from these data and how good data and information fulfil requirements that were recently proposed for disaster reporting on the European and international level. In accordance with the European Floods Directive, impacts on human health, economic activities (and assets), cultural heritage and the environment are described on the national and sub-national scale. Information from governmental reports is complemented by communications on traffic disruptions and surveys of flood-affected residents and companies.

Overall, the impacts of the flood event in 2013 were manifold. The study reveals that flood-affected residents suffered from a large range of impacts, among which mental health and supply problems were perceived more seriously than financial losses. The most frequent damage type among affected companies was business interruption. This demonstrates that the current scientific focus on direct (financial) damage is insufficient to describe the overall impacts and severity of flood events.

The case further demonstrates that procedures and standards for impact data collection in Germany are widely missing. Present impact data in Germany are fragmentary, heterogeneous, incomplete and difficult to access. In order to fulfil, for example, the monitoring and reporting requirements of the Sendai Framework for Disaster Risk Reduction 2015–2030 that was adopted in March 2015 in Sendai, Japan, more efforts on impact data collection are needed.

## 1 Introduction

In June 2013, large-scale flooding occurred in many Central European countries, i.e. in Switzerland, Austria, the Czech Republic, Slovakia, Poland, Hungary, Croatia,

Serbia, and particularly in Germany. In 45 % of the German river network peak flows exceeded the 5 year flood discharge (Schröter et al., 2015). Using an adapted method of Uhlemann et al. (2010) that determines and assesses large-scale flooding based on discharge data from 162 gauges from all over the country, the flood of June 2013 can be regarded – in hydrological terms – as the most severe flood in Germany over at least the past 60 years (Merz et al., 2014). However, the extreme flood of August 2002 remains the most damaging event with an overall loss of EUR 11.6 billion (as at July 2005; Thieken et al., 2006).

The event of 2013 was especially characterised by extraordinary high antecedent moisture. During the second half of May 2013 exceptional rainfall amounts had been witnessed due to a quasi-stationary upper-level trough over Central Europe. This circulation pattern triggered a sequence of surface lows on its eastern side, a process that was also referred to as repeated Rossby Wave Breaking (RWB; Grams et al., 2014) and that repeatedly transported warm and humid air from South-East Europe to Central Europe (Schröter et al., 2015). Notably continental evapotranspiration was the main moisture source as revealed by Grams et al. (2014). By the end of May, rainfall totalled to 178 % of the average monthly amount and record-breaking soil moisture was observed in 40 % of the German territory (DWD, 2013). Accordingly, Schröter et al. (2015) also reported high initial streamflow levels in the river network.

First local flooding was caused by a thunderstorm on 18 May 2013 in the southern part of Lower Saxony, where anew heavy rainfall and flooding occurred a week later (NLWKN, 2013). However, the large-scale flooding was mainly triggered by rainfall between 31 May and 2 June 2013. These rainfall amounts were considerable – especially over mountains – but not exceptional (Schröter et al., 2015). However, in combination with the wet soils and above-average initial streamflow levels, high flood peaks resulted in the upper catchments of the rivers Rhine and Weser in the Western part of Germany as well as in many parts of the catchments of the rivers Danube in Southern Germany and Elbe in East Germany.

**NHESSD**

doi:10.5194/nhess-2015-324

**The flood of June 2013 in Germany: how much do we know about its impacts?**

A. H. Thieken et al.

Flood discharges above a five-year return period were observed in many rivers reaches in Germany between 21 May 2013 and 20 June 2013. Over a length of approximately 1400 km in the river network even 100-year flood discharges were exceeded. Therefore, widespread inundation occurred as depicted in Fig. 1. At several locations, embankments were unable to withstand the floodwater resulting in dike breaches and inundation of the hinterland. Particularly affected areas are detailed in Fig. 1a–d, i.e. the areas inundated by a dike breach at Fischbeck at the river Elbe (Fig. 1a), at the confluence of the rivers Saale and Elbe at Klein Rosenburg-Breitenhagen (Fig. 1b) or in Deggendorf-Fischerdorf at the confluence of the rivers Isar and Danube (Fig. 1c). The city of Passau (Fig. 1d) is commonly known as "Three-River-City" since it is located at the confluences of the rivers Danube, Inn and Ilz. Due to its special geographic-topographic situation no flood defence schemes are in place. In 2013, the water level of 12.89 m above gauge zero nearly reached that of a flood event in 1501, which is with 13.20 m above gauge zero the highest water level ever recorded in Passau (BfG, 2013).

While the meteorological and hydrological aspects of the flood event were published in scientific journals already a few months after the flood (e.g. Blöschl et al., 2013; Grams et al., 2014; Merz et al., 2014; Schröter et al., 2015), only little information is available on the flood impacts. However, the societal significance of natural events such as floods only becomes visible through their effects on human society, its assets and activities. Accordingly, the crucial dimension when it comes to the assessment of events is not the flood hazard, but the flood risk. In this context, flood hazard is defined as the exceedance probability of potentially damaging flood situations and is often assessed by a frequency analysis of the discharges or the water levels at a given point within a specified period, usually a year (Merz and Thieken, 2004). Flood risk statements, in contrast, do include information about the consequences of flood situations, for example direct losses or fatalities. Hence, flood risks are not solely dependent on the flood hazard, but also on the vulnerability of the affected society. This is determined by the use of the flood-prone areas, i.e. the exposure of human

**NHESSD**

doi:10.5194/nhess-2015-324

The flood of June 2013 in Germany: how much do we know about its impacts?

A. H. Thieken et al.

beings, infrastructures and buildings to flooding (also referred to as elements at risk or damage potential), as well as the susceptibility of these elements to inundation. The extent of vulnerability and risk is strongly influenced by the resilience of the affected society or its ability to resist: the better the preventive and protective measures, early warning systems and emergency response have been developed, the less severe the resulting damage will be.

In general, adverse effects of floods are divided into direct and indirect damage (Smith and Ward, 1998). While direct damage, such as fatalities and injured people as well as damaged or destroyed buildings, are directly caused by a physical contact of the element at risk with the flood water, indirect damage occur in space and time outside the actual event. Among these effects are traffic and business disruptions, but also migration or long-term psychological illnesses.

Accounting for all impacts and costs of a particular event is complicated for many reasons (Downton and Pielke, 2005). To begin with, damage to buildings seems to be monetised easily since the goods concerned are traded on the market (Merz et al., 2010). Thus, the damage costs can be estimated on the basis of the necessary repair works and materials in a first instance. For some applications such as cost-benefit analyses, however, the financial damage that is based on repair and replacement costs has to be depreciated by the betterment that the damaged structures underwent during reconstruction; taxes also have to be excluded (see Merz et al., 2010). A monetary estimate can also be put on disruptions of operations, turnover losses or costs incurred by delivery detours. However, further indirect costs of disasters along production chains are difficult to measure and can often only be assessed by models (Greenberg et al., 2007; Meyer et al., 2013). Moreover, many losses (and benefits) associated with a flood event are intangible and difficult to monetarise or even to observe. Health effects due to flooding, but also damage to cultural heritage or the environment can only be monetised – if at all – through indirect assessments, based on for example the willingness of the population to pay for the restoration of a cultural heritage site or a recreational area, as well as to avert evacuation (see Meyer et al., 2013 for

**NHESSD**

doi:10.5194/nhess-2015-324

**The flood of June 2013 in Germany: how much do we know about its impacts?**

A. H. Thieken et al.

an overview). Furthermore, even big flood events have direct and indirect benefits, for example donations, relief funds or other (financial) support provided to affected regions, which should be crosschecked with the costs. Finally, flood losses might differ and depend on the spatial and temporal scale of the assessment, for example the property (asset), local, regional, national or international scale as defined by De Groeve et al. (2013), as well as on the overall context of the analysis and its underlying monetary assessment.

The true costs of flood events may hence include hidden costs, such as health effects and long term societal impacts, and hidden benefits caused by, e.g. extra compensation payments, which are difficult to identify and quantify (Downton and Pielke, 2005). Due to this complexity, there is currently a clear focus on accounting direct damage costs or primary effects of actual events (Pielke and Landsea, 1998) by using economic and/or human indicators (IRDR, 2015). While human indicators such as the number of people killed, injured or evacuated can be determined fairly reliable shortly after the event, a reliable estimate of the direct economic or financial costs of an event can often only be made after several years when all repair works and compensation payments have been completed. Using flood damage data provided by the National Weather Service (NWS) in the USA, Downton and Pielke (2005) demonstrated that reliable loss figures require regular data updates and consistent definitions of the damage components included. Data consistency is, however, difficult to assess if sub-amounts such as damage in different sectors or damage to movable and fixed items, are not explicitly recorded (Blong, 2004; Downton et al., 2005). Further potential biases of loss data are outlined by Gall et al. (2009).

In contrast to meteorology and hydrology, very little standardisation and institutionalisation prevails regarding (flood) loss documentation (Kreibich et al., 2014), although the lack of reliable, consistent and comparable data is seen as a major obstacle for effective and long-term loss prevention (Changnon, 2003). Enhanced efforts to collect loss data and the development of transparent methodologies and

**NHESSD**

doi:10.5194/nhess-2015-324

**The flood of June 2013 in Germany: how much do we know about its impacts?**

A. H. Thieken et al.

standardized datasets have been constantly demanded since an accurate, comparable and consistent impact database is required for many applications, among others:

- to assess the influences of climate, population growth, land use and policies on trends in losses and damage (Downton et al., 2005),

- to improve risk assessment methods by calibrating and validating loss models with real data (De Groeve et al., 2013),

- to identify drivers and root causes of disasters and to deepen our understanding of damaging processes (disaster forensic; DKKV, 2012),

- to set priorities between competing demands for national and international budget allocations (Guha-Sapir and Below, 2002),

- to evaluate policy successes and failures on the basis of trends and spatial patterns of damage,

- to think about new policies (insurance, climate policies),

- to set priorities of research funding, and

- to evaluate contributions of science to real-world outcomes (Downton and Pielke, 2005).

Since damage information is assumed to be collected more systematically and comprehensively for a major flood than for a small event and information is more likely to be shared among different agencies and institutions (Downton and Pielke, 2005), this paper explores what data is currently available to describe the impacts of the flood event of June 2013 and what can be learnt from them about the types and severities of flood impacts in different sectors. Finally, it will be discussed how good current data and information are and what could be done to create better impact data.

In consistency with the European Floods Directive (2007/60/EC) that aims to establish a framework for the assessment and management of flood risks in Europe

Discussion Paper | Discussion Paper | Discussion Paper | Discussion Paper | Discussion Paper |

**NHESSD**

doi:10.5194/nhess-2015-324

**The flood of June 2013 in Germany: how much do we know about its impacts?**

A. H. Thieken et al.

**NHESSD**

doi:10.5194/nhess-2015-324

**The flood of June 2013 in Germany: how much do we know about its impacts?**

A. H. Thieken et al.

and to reduce adverse consequences of flooding for human health, economic activities (and assets), cultural heritage, and the environment, predominantly data and impacts on these domains are explored in this paper. Further, we mainly concentrate on direct flood impacts on different scales, i.e. from the national down to the property (asset) scale, due to the above-mentioned complexities and problems that are associated with indirect and long-term effects. In the next section, the used data sources are introduced, before the actual flood impacts are presented per damage type and scale (if applicable) in Sect. 3. This part of the paper is accompanied by an overall evaluation of the data content and quality in comparison to recently published guidelines on recording disaster losses (Corbane et al., 2015; IRDR, 2015), which will be introduced in each section dealing with a damage category. The paper ends with recommendations on future event documentation and loss data collection.

## 2 Data sources

Three main data sources were used for this study: (i) governmental reports on the flood in June 2013, (ii) communications on disruptions of road and railway traffic, and (iii) computer-aided telephone interviews among flood-affected residents and companies.

### 2.1 Governmental reports

General information on the flood impacts was collected from official governmental reports on the flood on the federal/national level (e.g. BMF, 2013; BMI, 2013; BfG, 2014; GMLZ, 2014), as well as on the subnational level of the affected states (*Länder*; e.g. Saxon State Chancellery, 2013; Saxony-Anhalt Ministry of the Interior and Sport, 2013). In addition, enquiries on the overall losses detailed per economic sector and affected municipality were directed at the Federal Ministries of the Interior and of Finance as well as at the respective ministries of flood-affected states in spring

2014. All ministries responded; most of them referred to the numbers reported in the application of the German Federal Government to the European Union Solidarity Fund from July 2013 (BMF, 2013). Some states updated their loss estimates; almost none provided numbers on a finer spatial level. For Saxony, some numbers are documented
5  per administrative district (*Landkreis*) by the Saxon State Chancellery (2013); Bavaria reported costs for emergency services on the level of the seven Bavarian administrative regions (*Regierungsbezirke*; StMI, R. Schwab, personal communication, June 2014). The most recent numbers were published in a small parliamentarian enquiry (Federal Parliament, 2015).
In this paper, these governmental reports were used to retrieve information on the general human and economic indicators proposed by De Groeve et al. (2014); Corbane et al. (2015) and IRDR (2015) on the national and the subnational level. In addition, the reports provided insight into expenses for emergency services as well as into impacts on cultural heritage and the environment.

**2.2   Communications on disruptions of road and railway traffic**

Since the Floods Directive addresses impacts on economic activities, disruption of transportation plays an important role. Therefore, communications on the disruption of road and railway traffic were analysed.

**2.2.1   Road traffic**

In order to capture the impact of the June 2013 flood on road traffic systematically, all communications contained in police traffic reports for the period from 15 May to 31 December 2013 with respect to flooding were filtered out and the retrieved information was saved in a database. An example of a police traffic report with respect to the flood event reads as follows: "4 June 2013, 11:30 a.m.: B96 Hoyerswerda in the direction of
Bautzen, between junctions Zeissig and Neu Buchwalde traffic obstructions in both

Discussion Paper | Discussion Paper | Discussion Paper | Discussion Paper |

**NHESSD**

doi:10.5194/nhess-2015-324

**The flood of June 2013 in Germany: how much do we know about its impacts?**

A. H. Thieken et al.

directions due to flooding, traffic obstruction due to flood, both directions of traffic closed, a detour has been instated" (source: Saxon Police, 2013, own translation).

All situations that posed an obstruction to road traffic, such as a closed road on one side or on both sides, narrowing of lanes, obstructions by traffic (e.g. by emergency vehicles) as well as dangers (e.g. an increase in game crossing the road due to the flood) were further considered as traffic obstruction. Repeated identical reports were merged so that they counted as one traffic obstruction. However, should a piece of information in the report change, for example the stated section of the affected road, then the report was captured as a new traffic obstruction. A traffic obstruction was deemed to have ended, as soon as

- information in the report changed so that this could be captured as a new traffic obstruction,

- it had been reported that the street was traversable once again or that the danger on the road had passed,

- the traffic obstruction did not appear in the police traffic reports any longer.

### 2.2.2 Railway transportation

The German Railways Corporation (*Deutsche Bahn AG*; DB) provided several internal communication maps, in which the railway segments that were interfered due to extreme weather conditions or flooding are shown. The maps cover the time period between 3 June and 1 July 2015 with, however, some days without any information. On other days, especially at the beginning of the flood event, the maps were updated several times a day. Besides the geographic information, the type of interference, i.e. low-speed routes, platform or route closures, is reported in the maps. Further, the press releases of the DB were used to retrieve additional information.

**NHESSD**

doi:10.5194/nhess-2015-324

**The flood of June 2013 in Germany: how much do we know about its impacts?**

A. H. Thieken et al.

Discussion Paper | Discussion Paper | Discussion Paper | Discussion Paper | Discussion Paper |

**NHESSD**

doi:10.5194/nhess-2015-324

**The flood of June 2013 in Germany: how much do we know about its impacts?**

A. H. Thieken et al.

## 2.3 Computer-aided telephone interviews

To capture more detailed flood effects on the level of individual properties (assets, households), information from flood-affected residents and companies was systematically gathered.

### 2.3.1 Flood-affected residents

Computer-aided telephone interviews (CATI) were conducted among households in the flood-affected regions of Germany nine months after the event. On the basis of information from affected municipalities, flood reports or areas experiencing flooding, street lists were compiled and the telephone numbers of residents potentially affected by the flood were searched. For the survey on the 2013 event, a comprehensive survey was conducted, i.e. all the searched telephone numbers were contacted. In total, 1652 interviews were completed between 18 February and 24 March 2014 with affected households. In the survey, the term "affected" was defined as a household that had suffered (financial) flood damage in May or June 2013.

Similarly to former surveys (see Thieken et al., 2005; Kienzler et al., 2015), the main objective was to investigate how financial flood losses are influenced by other factors, for example flood characteristics or private mitigation. However, after the June 2013 flood, some questions were posed regarding flood effects on health and wellbeing as well as on the assessment of the (governmental) aid for reconstruction. Overall, the questionnaire addressed the following topics (in the order of appearance):

- hydraulic characteristics of the flood at or in the building;

- early warning and emergency measures;

- contamination of the floodwater;

- evacuation;

- clean-up work and recovery;

- adverse flood effects, including effects on health and wellbeing, and perceived severity;

- physical and financial flood damage to the building and the household contents;

- building ownership and further information on the residential building (or the rented apartment);

- previously experienced flood events and flood awareness;

- long-term preventive and protective measures undertaken by the affected household and motivation (not) to do so;

- aid and financial compensation;

- socio-demographic information.

Information on health effects and the perceived severity of different damage types presented in Sect. 3.1.2 of this paper are based on this survey.

The above-mentioned surveys that were conducted a few months after the floods in 2002, 2005 and 2006 (see Kienzler et al., 2015) were complemented by a follow-up household survey in autumn 2012 ($n = 910$ households), i.e. ten years after the flood in 2002. The survey follow-up focused on long-term (health) effects of the floods as well as property-level mitigation measures. These data are used in Sect. 3.1.2 to illustrate short and long-term flood effects on affected residents.

## 2.3.2 Flood-affected companies

Companies that had been affected by the flood in June 2013 were surveyed with regard to the losses incurred and the circumstances influencing the type and amount of damage. For the sampling procedure, street lists were compiled on the basis of information obtained from municipalities, flood reports or mapped inundation areas and were further used to determine the telephone numbers of companies potentially

Discussion Paper | Discussion Paper | Discussion Paper | Discussion Paper | Discussion Paper |

**NHESSD**

doi:10.5194/nhess-2015-324

**The flood of June 2013 in Germany: how much do we know about its impacts?**

A. H. Thieken et al.

affected by the flood. To include some large-sized companies in the random sampling as well, these were searched additionally from flood reports.

Affected companies were surveyed from mid of May to mid of July 2014. Again, the term "affected" was defined as an enterprise that had suffered (financial) flood damage. The information was gathered through CATI with the individual in the company who was best placed with providing information on the flood. In total, 557 interviews were completed. The interviews lasted 15 to 35 min on average; the questionnaire covered approx. 90 questions on the following topics (in the order of appearance):

– company description (sector, size, number of buildings, assets, perceived vulnerability with regard to flooding, etc.);

– hydraulic characteristics of the flood on the company grounds;

– early warning and emergency measures;

– contamination and clean-up work;

– (financial) flood damage (to buildings, operational facilities, merchandise, products and warehouse inventory, motor vehicle inventory; due to interruptions of operations);

– reconstruction, compensation, plans to relocate;

– previously experienced floods;

– long-term preventive and protective measures at the property-level.

Results presented in Sect. 3.4 of this paper are based on this survey.

**NHESSD**

doi:10.5194/nhess-2015-324

**The flood of June 2013 in Germany: how much do we know about its impacts?**

A. H. Thieken et al.

Discussion Paper | Discussion Paper | Discussion Paper | Discussion Paper |

Discussion Paper | Discussion Paper | Discussion Paper | Discussion Paper | Discussion Paper |

**NHESSD**

doi:10.5194/nhess-2015-324

**The flood of June 2013 in Germany: how much do we know about its impacts?**

A. H. Thieken et al.

## 3 Impacts of the flood in June 2013

### 3.1 Flood impacts on human health

The effects of flooding on health can be significant and may concern both, physical and mental health. Physical health effects are deaths due to drowning, electrocution, heart attacks, vehicle-related accidents etc. as well as injuries, illnesses and infections that require medical assistance and result directly from the flood, for example due to a lack of sanitation, contaminated water, chemical hazards or mildew (within wet or insufficiently reconstructed buildings; IRDR, 2015). Mental health effects might be acute or long-term due to a loss of family members or friends, displacement, destruction of homes, delayed recovery and water shortages (Menne and Murray, 2013). Recurrent flash backs, nightmares, sleeplessness (insomnia), angst, panic and depression are some examples for mental health effects and might even lead to a posttraumatic stress disorder (PTSD). Limited access to health facilities during and after a flood event, in particular medical treatment and nursing of flood-affected or evacuated people suffering from chronical diseases, is a further issue related to this domain (Menne and Murray, 2013), but this is usually reported as part of the physical and economic damage (see Sect. 3.2).

Human indicators in disaster loss databases are commonly related to physical health or the displacement and movements of people caused by the flooding; mental health effects are usually not explicitly reported. For example, IRDR (2015) proposes the numbers of dead, missing, injured and exposed people as primary human impact indicators, while the numbers of homeless, evacuated, relocated and affected people are regarded as secondary. Some indicators, e.g. dead and missed people, are mutual exclusive, others, e.g. homeless, evacuated and relocated people, are not since they correspond to consecutive management phases of a damaging event (IRDR, 2015).

In the European guidance for recording disaster losses (Corbane et al., 2015), the number of deaths, missing people as well as directly affected people are recommended as minimum information that should be recorded with regard to human losses. All

information should be provided on the NUTS 2 or NUTS 3 level (NUTS stands for Nomenclature of Territorial Units for Statistics). In Germany, these mainly correspond to the 38 (former) administrative regions (*Regierungsbezirke)* as well as 402 urban and rural administrative districts (*kreisfreie Städte und Landkreise*), respectively. Since

almost no information was provided for administrative levels below the federal states, an overview of human loss indicators (see Table 1) can currently be provided for this level (NUTS 1) only.

### 3.1.1 Overview of human loss indicators for the flood in June 2013

Table 1 illustrates that 14 people lost their lives in the June 2013 flood. Five fatalities

can be allocated to Saxony-Anhalt (Saxony-Anhalt Ministry for the Interior and Sport, 2013), three to Baden-Wurttemberg (Die Welt, 2013), two to Saxony (BMF, 2013) and one to Bavaria (BMF, 2013). In fact, in the application of the German Government to the European Union Solidarity Fund, only eight (immediate) fatalities were reported. This number was later corrected to 14 (GMLZ, 2014). In addition, 128 people were injured

and approximately 80 630 were evacuated in eight different federal states (GMLZ, 2014). In general, 600 000 people in 1800 municipalities were affected by the flood (BMF, 2013; Table 1). However, the term "affected" is not clearly defined, nor is its relation to the categories "injured" and "evacuated" in terms of ex-/inclusiveness. Due to this ambiguity, IRDR (2015) recommends using exposed people, defined as the

number of people who permanently or temporarily reside in the hazard area before or during the event, in a first instance, as this number can be more reliably determined from census data and geographic information on the flooded area. So far, the number of people exposed to the June 2013 flood has not been determined.

As a further human-related indicator, the number of helpers in emergency services

and relief or aid organisations totalling to more than 1 million person days Germany was often reported in governmental reports dealing with the June 2013 flood and hence added to Table 1 although this indicator is not considered in any guideline for disaster documentation. This number does not include volunteers who helped to cope with the

**NHESSD**

doi:10.5194/nhess-2015-324

**The flood of June 2013 in Germany: how much do we know about its impacts?**

A. H. Thieken et al.

Discussion Paper | Discussion Paper | Discussion Paper | Discussion Paper

flood (damage) without being organised in an emergency service or an aid or relief organisation. With regard to the 2013 flood, the numbers illustrate that the magnitude of responders and helpers is similar to the amount of people directly affected by this widespread flood event.

Table 1 clearly demonstrates that the reporting of the federal states was not focussed on human losses. Only in the Bavarian report (Annex 6 in BMF, 2013), numbers for all categories of Table 1 were mentioned. In most of the states, no (accessible) numbers were reported. Therefore, more transparent and systematic reporting procedures are needed to evaluate the quality of the aggregated data and to reach a comprehensive report on human losses that fulfils the minimum requirements proposed by Corbane et al. (2015) or IRDR (2015).

### 3.1.2 Flood impacts on affected residents and perceived severity

To obtain more insights of the variety and severity of flood impacts on affected residents, the surveyed households (see Sect. 2.3.1) were asked to indicate from a list of ten possible flood damages, which of these had affected them in June 2013 and how seriously they perceived each of the witnessed damage type. The answers could be graded on a scale of 1 (= damage was not serious at all) to 6 (= damage was very serious). The perceptions of all respondents to a particular damage type resulted in average assessments between 3.0 and 4.6 (Fig. 2). The damage types that were assessed on average with 4.0 or worse – and were thus evaluated as serious – cited as a priority and included psychological stress or other stresses, reinstatement works (e.g. cleaning or repairs), supply problems (e.g. no electricity, water etc.) as well as damage to buildings and household contents (Fig. 2). This highlights that mental health issues and disruption of daily life are of great importance for affected people.

Figure 2 further elucidates that the flood situation did not only have a great impact on the mental health of the affected persons, but also – to a lesser degree – on their physical health. To shed some light on the underlying medical conditions, all respondents who had reported mental or physical health effects (84.4 %) were

**NHESSD**

doi:10.5194/nhess-2015-324

**The flood of June 2013 in Germany: how much do we know about its impacts?**

A. H. Thieken et al.

Discussion Paper | Discussion Paper | Discussion Paper | Discussion Paper

surveyed in more detail. In an open question regarding the type of stress undergone and grievances in detail, uncertainty about the future, worries with regard to family, existence and subsistence, and the future, fears of loss, panic, trauma, shock, crying fits or nervous breakdowns were cited most frequently. In addition to these, sleep
disorders or nightmares were mentioned, as well as feeling restless, tense and nervous or agitated.

Physical symptoms manifested themselves most frequently in the form of states of exhaustion or lack of sleep; joint, bone, muscle or nervous complaints; infections, inflammation, (skin) irritations or the exacerbation of pre-existing illnesses or
conditions. It is noteworthy that the management of the flood situation aggravated in the case of persons with chronic illnesses or conditions.

Psychological stress is, however, not limited to the period of the actual flood event, but can still remain in existence a long time afterwards. The above-mentioned surveyed group of affected persons with health impairments was therefore additionally asked
about the extent to which they were still stressed by the flood event at the point in time of the interview (answer scale from 1 = "I am not stressed by it any longer/I feel like I did before the event" to 6 = "I am still very stressed by it"). Slightly more than a third of the respondents (35 %) were still very or extremely stressed as a result of the flood as much as nine months after the event (answers 5 and 6); by contrast, a further third
hardly felt stressed any longer or not at all (answers 1 and 2).

However, the 2013 flood was still very prevalent in the minds of all the residents affected. This is clearly evident from the results to the question: "How often have you thought about the June 2013 flood over the past six months?" At the point in time of the survey, i.e. approximately nine months after the event, 35 % of all the affected persons
still thought about the 2013 flood once or several times a day, 50 % still at least once a month to several times a week (Fig. 3). This distribution of answers clearly differs from the answers of affected persons who were asked the same question in autumn 2012, i.e. ten years after having witnessed the severe flood of August 2002 (Fig. 3). On the one hand the comparison illustrates the extent to which a flood can change

**NHESSD**

doi:10.5194/nhess-2015-324

**The flood of June 2013 in Germany: how much do we know about its impacts?**

A. H. Thieken et al.

daily life and thinking, on the other hand the long-term and ongoing impression that an extreme flood can leave behind is evidenced: ten years after the event of August 2002, only 20 % of respondents stated that they never thought of the event in the six months preceding the interview. 8 % still thought of it approximately daily. However, it is worth noting that the thoughts about the flood were not negative throughout: the experience of solidarity and a sense of community were often positively highlighted.

According to Kuhlicke et al. (2014), affected households in Saxony that had been flooded up to three times in recent years (i.e. in 2002, 2006 or 2010, and 2013), perceived the flood impacts more severe than households that had been affected by flooding in 2013 for the first time. In addition, households that already suffered flood damage several times thought considerably more often about resettlement which might have severe consequences for flood-prone communities that do not get flood protection (Kuhlicke et al., 2014).

In conclusion, the survey among flood-affected residents highlights the importance of physical and particularly mental health issues caused by flooding. This is contrasted by the little attention this domain received in official governmental flood documentations and reports in Germany.

## 3.2 Overview of impacts on economic activities (and assets) on the regional and national scale

In industrialised countries, economic or financial losses caused by natural hazards are a major concern and achieve a lot of attention during and after disastrous events. Quick and reliable loss estimates are requested by the (re-)insurance industry as well as by governmental institutions. However, data on economic or financial losses are fairly uncertain (Merz et al., 2004; Downton and Pielke, 2005; Downton et al., 2005). To assess impacts of natural hazards on the economic activities according to the European guidance for recording disaster losses (Corbane et al., 2015), indicators describing the physical number of damaged items should be distinguished from indicators that quantify financial loss (costs). As a minimum requirement, it is proposed

**NHESSD**

doi:10.5194/nhess-2015-324

**The flood of June 2013 in Germany: how much do we know about its impacts?**

A. H. Thieken et al.

**NHESSD**

doi:10.5194/nhess-2015-324

**The flood of June 2013 in Germany: how much do we know about its impacts?**

A. H. Thieken et al.

that physical damage indicators should deliver information on the number of damaged or destroyed houses, educational centres (e.g. schools, kindergartens) and health facilities (e.g. hospitals). Optionally, further aggregated damage indicators can be provided, i.e. on the total area of destroyed or affected crops and woods (in hectares), the number of lost four-legged livestock, the number of damaged or destroyed governmental and administrative buildings, the number of damaged or destroyed industrial and commercial facilities as well as the length of damaged or destroyed roads and railways (in kilometres) and the number of damaged or destroyed transportation infrastructure such as bridges, airports and marine ports (Corbane et al., 2015). These physical damage indicators are further translated into economic monetary indicators, in particular into the overall direct tangible loss, i.e. the monetary value of the physical damage to capital assets. This loss should ideally be disaggregated over all sectors or loss owners and accompanied by information on the loss bearer. Expenditures for emergency services and clean-up are further costs to be recorded optionally (Corbane et al., 2015).

As a minimum requirement the overall direct damage should be reported on NUTS 2- or NUTS 3-levels (see Sect. 3.1). As outlined above, almost no information was provided for administrative levels below the state-level. Therefore, the overview of damage and losses can currently be provided for the NUTS 1-level only. Table 2 summarises the information that was collected for the flood of June 2013 on the minimum indicators on direct damage and economic loss as proposed by Corbane et al. (2015).

### 3.2.1 Overview of financial losses

Twelve out of 16 federal states were affected by flooding between 18 May and 4 July 2013; in parts of eight federal states a state of emergency was declared (BMI, 2013). Table 2 illustrates that data on the physical damage indicators are so fragmentary that they do not allow a sound interpretation. Only from the Saxon report (Annex 14 in BMF, 2013; Saxon State Chancellery, 2013), information for all minimum

**NHESSD**

doi:10.5194/nhess-2015-324

**The flood of June 2013 in Germany: how much do we know about its impacts?**

A. H. Thieken et al.

indicators recommended by Corbane et al. (2015) could be retrieved. Therefore, the overall (direct) financial loss given in Table 2 is further used as main indicator for the economic impact.

According to the Federal Ministry of Finance (BMF, 2013), the overall losses that incurred in the June 2013 flood amounted to EUR 8154 million. This figure was communicated by the Federal Government in its application to the European Union Solidarity Fund mid of July 2013. The answers to our queries in spring 2014 (see Sect. 2.1) indicate that this estimate will clearly turn out to be less than had initially been anticipated. The two most significant corrections were communicated by Saxony-Anhalt and the Federal Government. The loss in Saxony-Anhalt, originally estimated at EUR 2.699 billion, was reduced to between EUR 1.5 and 2 billion (written communication from the Saxony-Anhalt Ministry of Finance dated 15 April 2014). In addition, the damage to the infrastructure of the Federal Government – this involves damage to the federal assets regarding railways, motorways and navigable waterways, as well as to the administrative buildings of the Federal Government – clearly lies below the EUR 1.484 billion estimated initially (Table 2). In its response to a small parliamentary enquiry concerning the flood relief funds, the Federal Government recently assumed that only a sum of approximately EUR 114 million was in question (Federal Parliament, 2015). The same enquiry also provides recent amounts of losses that have been claimed to governmental relief funds by the end of June 2015 (see Table 2). In contrast to the application to the European Union Solidarity Fund, these numbers, however, seem not to include expenses for emergency response, nor insured losses. Losses that property owners bear themselves are probably also neglected in these figures. With these corrections and considerations, the total direct loss will probably not exceed EUR 6 billion.

Even the most recent numbers indicate that the compilation of the overall financial losses is still preliminary. Many of the damage claims have not been resolved conclusively and to some extent unforeseeable losses that had been incurred but have not been reported may still appear. According to the administrative arrangement for the

Act to Establish Reconstruction Funds passed in 2013, applications for reconstruction aid could be submitted until 30 June 2015. The period for final approval was recently extended to 30 June 2016 (Federal Parliament, 2015). Only thereafter will it be possible to compile a conclusive loss statement.

Nevertheless, it is already possible to look at the spatial and sector-wise distribution of losses. Table 2 reveals that Saxony-Anhalt, Saxony and Bavaria are the three most affected federal states in terms of financial losses, each covering about 20 to more than 30 % of the overall loss. For the flood of 2013, approximately 22 % of all losses incurred in private households, 19 % in the industrial and commercial sector, 7 % in agricultural and forestry and almost 50 % in governmental domains (infrastructure and emergency services; BMF, 2013). This distribution can, however, considerably vary between federal states as is illustrated in Fig. 4 taking Bavaria and Saxony as examples. While in Bavaria two thirds of the losses are allocated to private households as well as the commercial and industrial sectors, losses to the state and municipal infrastructure amount to around 60 % in Saxony (Fig. 4). This can be divided into 20 % state infrastructure and 40 % municipal infrastructure. In the area of state infrastructure, the biggest damage can be attributed to surface water bodies and flood defence systems belonging to Water Body Category I. With regard to municipal infrastructure, the largest share of the damage is allocated to streets and bridges, as well as to flood defence systems belonging to Water Body Category II (Saxon State Chancellery, 2013).

It is noteworthy that flood losses in Germany are generally divided into the sectors private households, industry and commerce, agriculture and forestry, state and municipal infrastructure as well as costs for emergency services in loss statements of the Federal Government and the federal states. To some extent, losses to cultural facilities, sport and recreational centres, churches and research institutions are also provided. Unfortunately not all the sectors are systematically dealt with in every flood event and state, and the definitions, which damage should be reported in which category, are not managed uniformly over space and time. Moreover, in the case of

**NHESSD**

doi:10.5194/nhess-2015-324

**The flood of June 2013 in Germany: how much do we know about its impacts?**

A. H. Thieken et al.

changes to the overall loss estimates, the distribution among the sectors is often not updated. This considerably hinders a comparison of the overall financial or economic losses of different flood events and in different federal states (Thieken et al., 2010).

Of all losses, the insurance industry in Germany has covered around EUR 1.65 billion (GDV, 2015). With EUR 900 million, the most insured damages occurred in the Freestate of Saxony, followed by Saxony-Anhalt (EUR 310 million), Bavaria (EUR 270 million) and Thuringia (EUR 140 million; GDV, 2014). 142 major claims – this equals to individual claims exceeding EUR 500 000 – were reported to the Association of the German Insurance Industry (GDV) with an overall damage total of EUR 257 million (GDV, 2015).

In addition, the Federal Government and all federal states launched flood relief funds containing a total amount of EUR 8 billion. The parties agreed that losses of private households can be compensated up to 80 %, whereas repair costs for damaged state and municipal infrastructure can be covered up to 100 %. Further, private donations of EUR 108 million have been available (BMF, 2013). Altogether, the funds available for reconstruction excel the total damage. Therefore, more reliable methods for first and immediate damage estimates are required. In order to evaluate the reasonability of first loss estimates reported by the federal states to the Federal Government, not only the estimation methods applied should be documented, but the numbers of physically damaged (or destroyed) items should also be reported by default as is suggested by loss data guidelines (e.g. Corbane et al., 2015). For a first estimate, the number of damaged or destroyed items could be combined with standard repair costs per item. Further, damage indicators should be clearly defined and agreed upon so that the loss documentation of different states and events can be better compared.

### 3.2.2 Expenses for disaster response and emergency services

With the 2013 flood situation, distinctive needs arose for disaster response and appropriate support by personnel and technical resources in the affected federal states. While Baden-Wuerttemberg and Bavaria managed the flood situation predominantly

**NHESSD**

doi:10.5194/nhess-2015-324

**The flood of June 2013 in Germany: how much do we know about its impacts?**

A. H. Thieken et al.

**NHESSD**

doi:10.5194/nhess-2015-324

**The flood of June 2013 in Germany: how much do we know about its impacts?**

A. H. Thieken et al.

with their own teams and resources, as well as via bilateral cooperation, states like Saxony, Thuringia and Saxony-Anhalt used the coordination service offered by the German Federal Joint Information and Situation Centre (GMLZ). Aside from support by the federal states not affected, the states worst affected also organised themselves into providing joint support for one another according to the Federal Ministry of the Interior (BMI, 2013).

The assistance requests of individual federal states had already been brought to the attention of the GMLZ on 2 June 2013. In total, the GMLZ processed 43 assistance requests from five affected states. Subsequently, around 5.15 million sandbags, 5700 emergency rescue personnel and transport services for 1000 tonnes of material were arranged in the course of the flood situation by 15 June 2013 (GMLZ, 2014). Material shortages occurred in the number of available sandbags. To meet the demand of the affected areas, the GMLZ arranged for five million sandbags from other federal states and Germany's European neighbours (BMI, 2013).

In total, 1.7 million voluntary workers are organized in (volunteer) fire brigades, relief and aid organisations as well as the German Agency for Technical Relief (THW). They form the cornerstone of Germany's disaster response. By 5 July 2013, the deployment of local fire brigades and aid organisations added up to around 871 000 person days (GMLZ, 2014; Table 1). Additionally, the Federal Government supported affected municipalities and states with its own resources. In the process, the Ministry of the Interior (BMI) coordinated the support staff of the Federal Police and the THW, while the Ministry of Defence coordinated the Federal Armed Forces staff. From the outset of deployment, the Federal Government provided help in the form of around 216 000 person days (GMLZ, 2014; Table 1). Through this, the Federal Government incurred additional costs to the tune of EUR 59.9 million (BMF, 2013).

In general, the costs for emergency services and response are included in the overall loss estimates shown in Table 2. Three federal states explicitly reported their response costs, which amount to EUR 8.89 million in Bavaria (by 25 June 2014), EUR 1.70 million in Schleswig-Holstein and EUR 0.99 million in Thuringia. Related to

the total amount of the other direct damage costs as at July 2015, the response costs of these federal states amount to 1.2, 12.1 and 0.5 % of the direct damage, respectively, which considerably differs from the 2 % of the direct damage that is often used to estimate response costs ex-ante (see Penning-Rowsell and Wilson, 2006;
Pfurtscheller and Thieken, 2013). Since publicly accessible data in this domain is scarce despite well-established costing and reporting procedures, explicitly reporting of costs for emergency services and disaster response in loss documentations is highly recommended.

### 3.3 Impacts on economic activities – traffic disruptions

Apart from the direct damage to assets presented in Sect. 3.2, floods can have further adverse impacts on economic activities – also far beyond the flooded area – for example if the transportation systems are affected. In general, roads, railways, waterways as well as airports play an important role for the transportation of goods and people. Therefore, traffic disruptions during the flood in June 2013 are analysed
in this section although in the guidelines on loss documentation (Corbane et al., 2015) damage to the transportation system is only considered in terms of physical damage (see Sect. 3.2). Traffic disruption include complete interruption of operations due to route closures as well as restrictions to normal operations on damaged routes, for example: on dual track/carriage routes only one track/lane is usable, low-speed routes
or diverted routes are implemented, or the transportation system is replaced by another mode of transport (e.g. railways are replaced by buses).

In the longer term, disruption of a particular mode of transport might lead to a loss of customers or a decline in customer satisfaction, for example with railway services. Such effects of flood events are, however, difficult to separate from other influencing
factors. Therefore, the analysis focusses on traffic disruptions and interferences.

**NHESSD**

doi:10.5194/nhess-2015-324

**The flood of June 2013 in Germany: how much do we know about its impacts?**

A. H. Thieken et al.

Discussion Paper | Discussion Paper | Discussion Paper | Discussion Paper

### 3.3.1 Disruption of navigation

If rivers are used as waterways, river reaches will be closed for navigation when a specified water level, i.e. the highest navigable water level, is exceeded at the respective reference gauge. As summarized in Table 3, such water levels were observed in June 2013 at several gauges on different Federal waterways and lasted for 15 consecutive days at maximum. The internationally important waterway at the Lower Rhine was, however, not affected by this flood (BfG, 2014).

Disruption of the shipping traffic might last longer than the durations given in Table 3, since the Federal Waterways and Shipping Administration first has to screen for new obstacles in the navigation channels before these can be regularly navigated again.

So far, no monetary assessments of the disruption of waterways have been undertaken (BfG, 2014). Related costs are therefore not included in the figures of Table 2.

### 3.3.2 Disruption of road traffic

The flood event of 2013 led to flooding, dangerous situations and closures of streets in city centres, closures of regional roads and even of a Federal motorway (*Autobahn*). In total, 700 km of roads and 150 bridges were damaged in Germany (BMF, 2013). These impacts resulted in interferences of road traffic across almost the whole of Germany.

The chronological sequence of traffic obstructions on German roads is illustrated in Fig. 5 and reflects the general development of the flood as described by Schröter (2015; see also Sect. 1). Isolated reports of flood-related traffic obstructions emerged as early as 19 May 2013. As of 26 May, the flooding of the rivers Weser and Leine, particularly in the administrative districts of Braunschweig and in the Hanover region in Lower Saxony, was the presumed reason behind road closures. As of 31 May, numerous traffic obstructions occurred in almost all of the federal states, especially in the most affected, i.e. Bavaria, Saxony and Saxony-Anhalt. On 2 June 2013, traffic obstructions had reached a maximum nationwide (Fig. 5). Due to the flood developing over several

**NHESSD**

doi:10.5194/nhess-2015-324

**The flood of June 2013 in Germany: how much do we know about its impacts?**

A. H. Thieken et al.

days, it was only as of 6 June 2013 that traffic obstructions from the flood occurred at the lower reaches of the River Elbe.

More than 75 % of reported traffic obstructions can be traced back to the actual flooding of streets or to flood danger (Fig. 5). In addition to these, landslides especially in Baden-Wuerttemberg (Keller and Atzl, 2014) together with numerous uprooted trees contributed to approximately 20 % of obstructions in road traffic. In more than 60 % of the events, the roads had to be closed completely in both directions. Of the traffic obstructions, 10 % occurred in city centres and on other urban roads. The federal trans-regional road network was affected by more than 50 % of the traffic obstructions.

Figure 6 illustrates the spatial distribution and duration of the traffic obstructions on an administrative district level. With traffic obstructions lasting more than 14 500 h in total, traffic in Saxony was the most curtailed. The Saxon administrative districts of Meissen, Leipzig district, city of Dresden, Saxony's Swiss-East Ore Mountains and Central Saxony were equally affected by a very high incidence of traffic obstructions, as was the Hanover administrative region in Lower Saxony. However, it took only days to remove most of these after the flood had been cleared. In the administrative districts of Traunstein (Bavaria) and Tuebingen (Baden-Wuerttemberg) extensive construction work to damaged roads had to be conducted, which to some extent still affected regional traffic months afterwards.

The administrative districts denoted in red in Fig. 6 therefore all display a high overall duration of traffic obstruction. This information does not, however, indicate any decisive conclusions arrived at as to the actual indirect cost due to detours etc. incurred.

### 3.3.3 Disruption of railway operations

One company that has been considerably affected by the flood event of 2013 is the German Railways Corporation (*Deutsche Bahn AG*). In June 2013, mudslides as well as the submergence or under-washing of tracks led to a variety of interferences of the normal rail traffic (Fig. 7). Thus the morning of 3 June 2013 saw 60 route closures and interferences, of which approximately 25 were in Bavaria as well as approximately

Discussion Paper | Discussion Paper | Discussion Paper | Discussion Paper |

**NHESSD**

doi:10.5194/nhess-2015-324

**The flood of June 2013 in Germany: how much do we know about its impacts?**

A. H. Thieken et al.

30 in Thuringia and Saxony. In the afternoon, further restrictions were reported on up to 15 routes. These could be lifted to some extent in the subsequent days. From 8 June 2013, when the flood attained the middle reaches of the River Elbe, this number increased to 17 routes.

In the medium term, primarily long-distance traffic had to bear the brunt of the flood after the dyke breach at Fischbeck on 10 June 2013 (Fig. 1a) resulted in the flooding of an approximately 5 km long stretch at the town of Stendal. This meant that the high-speed rail line between Berlin and Hanover had to be interrupted until 4 November 2013, i.e. for almost five months (Deutsche Bahn, 2013). For this reason, important connections between Berlin and the Ruhr district, Cologne and Bonn, as well as between Berlin and Frankfurt (on the Main) were affected. A replacement timetable with diversions was deployed but led to travel time extensions of between 30 and 60 min (Deutsche Bahn, 2013). As a result, approximately 10 000 passenger trains and more than 3000 goods trains had to be diverted (Deutsche Bahn, 2014). Due to the travel time extension, a third of passengers took a flight to or from Berlin, or continued their journey by car or intercity coach (Deutsche Bahn, 2014). The financial impacts of this disruption on the railway company itself and on further economic activities are difficult to evaluate and are hence not included in the numbers presented in Table 2.

### 3.4 Impacts on economic activities at the asset scale: with a focus on business interruption of individual companies

Although the EU Floods Directive explicitly addresses the effects flooding has on economic ACTIVITIES, current loss guidelines and reporting emphasise adverse effects on ASSETS. In order to further complement the nationwide data that was presented in Sect. 3.2 and focused on losses to assets, this section looks at the diverse impacts floods can have on individual companies. The data from the survey described in Sect. 2.3.2 were used as basis for the analysis.

Flood impacts on companies comprise direct damage to buildings or merchandise, losses due to operational disruptions as well as indirect damage caused by delivery

**NHESSD**

doi:10.5194/nhess-2015-324

**The flood of June 2013 in Germany: how much do we know about its impacts?**

A. H. Thieken et al.

Discussion Paper | Discussion Paper | Discussion Paper | Discussion Paper

difficulties of suppliers (Fig. 8). Most of the companies surveyed, i.e. 88 %, indicated that they had been affected by operational disruptions (Fig. 8). This led to a similarly large percentage of turnover losses. Thus losses due to business interruption might be equally important than direct asset losses. Except for the amount on insured losses,
they are not included in the overall losses given in Table 2 since they are more difficult to assess than repair costs.

In general, different methods are available for the estimation of business interruption costs. The most prevalent approaches are (1) to apply a sector-specific reference value per unit affected or per day of interruption to estimate the loss of added value, (2) to
compare production output between hazard and non-hazard years, and (3) to calculate production losses as a fixed share of direct damages (Meyer et al., 2013). Since the first approach is the most reliable, the companies surveyed about the 2013 flood were further asked about the period of interruption of operations in their company, as well as how long it took for normal operations without any restrictions to resume afterwards
(period of restricted operations). The median value of downtime, through complete interruption of operations or restriction of operations, accounted for two to eight weeks, respectively. In the case of the 2013 flood, there were, however, a number of companies that experienced far longer downtimes through interruption of operations or restrictions of operations: the 75 %-percentile of downtime through the interruption of operations
due to the 2013 flood amounts to 60 days, the duration with restrictions of operation to 150 days. The average loss caused by interruption (or restriction) of operations amounted to EUR 137 287 ($n = 358$; median: EUR 15 000). They only exceeded the losses due to damaged equipment or buildings (see Table 4). Significant differences might, however, occur between different economic sectors, both in terms of downtimes
through interruption of operations and in terms of loss share as was shown for the 2002 flood by Kreibich et al. (2007).

**NHESSD**

doi:10.5194/nhess-2015-324

**The flood of June 2013 in Germany: how much do we know about its impacts?**

A. H. Thieken et al.

### 3.5 Impacts on cultural heritage

Although research and data on flood impacts predominantly deal with the impacts on economic assets, the European Floods Directive also addresses effects on cultural heritage. In fact, the flood in August 2002 severely damaged, for example the historic Semper opera house in Dresden (Saxony) and the Garden Kingdom in Dessau-Woerlitz (Saxony-Anhalt) approved as UNESCO world heritage and almost destroyed the flower gardens of the castle Weesenstein at the river Müglitz (Saxony; see DKKV, 2003). Since the repair and reconstruction work of such assets is often very specific and sometimes undoable and since the value that people attribute to such places is beyond financial accounting, this category is treated differently in the European guidance for recording disaster losses: not necessarily financial losses, but lists of damaged cultural, historical and UNESCO world heritage assets are proposed as indicators for loss databases (Corbane et al., 2015).

Despite the difficulties of the monetarization of damages to cultural heritage, financial losses related to cultural assets are given in the report of the German government to the European Union Solidarity Fund. The overall amount of initially EUR 56 million (BMF, 2013) accounts for only 1 % of the overall financial losses listed in Table 2. Saxony-Anhalt, Thuringia and Saxony reported losses to cultural assets of more than EUR 10 million each, Bavaria more than EUR 6 million, Baden-Wuerttemberg around EUR 1 million and Schleswig-Holstein EUR 350 000. A detailed list of affected cultural, historic or heritage assets is, however, missing. Thuringia explicitly mentions several damages to parks and gardens, for example the historic Greizer Landscape Park, a cultural place of national importance (Annex 16 in BMF, 2013). Schleswig-Holstein mentions inundation of the historic and listed centre of the city of Lauenburg on the Elbe. Furthermore, it is known that the (historic) city centres of Passau (Bavaria) as well as Grimma and Meißen (Saxony) were flooded. In contrast to 2002, the Garden Kingdom in Dessau-Woerlitz (Saxony-Anhalt) was rarely inundated despite higher water levels due to the meantime betterment of the flood protection. Nevertheless,

Discussion Paper | Discussion Paper | Discussion Paper | Discussion Paper |

**NHESSD**

doi:10.5194/nhess-2015-324

**The flood of June 2013 in Germany: how much do we know about its impacts?**

A. H. Thieken et al.

rising groundwater damaged one castle of this UNESCO world heritage site so that the restoration of the Garden Kingdom is with more than EUR 22 million one of the most expensive projects of the governmental relief funds (Federal Parliament, 2015). This demonstrates the high importance cultural heritage might have in individual cases.

## 3.6 Environmental impacts

Similar to the impacts on cultural heritage, it is difficult to quantify flood impacts on the environment (see Meyer et al., 2013). In fact, some impacts that are adverse at first sight might be ambiguous due to the fact that floods are natural phenomena and ecosystems in floodplains are adapted to flooding. Nevertheless, the environment can be damaged, especially due to inorganic and organic harmful substances that have dissolved or are transported with sediments and floodwater and enter freshwater systems. With regard to contaminants that are deposited in meadows, pastures and agriculturally used lands there is the risk of organisms absorbing them so that the contamination may sustain in food chains. Therefore, pollution must be regarded as the main indicator of adverse environmental flood impacts. Flooded protected ecosystem habitats and formation of new water bodies are further items that are considered by the European guidance for recording disaster losses (Corbane et al., 2015).

Adverse environmental effects might also occur, if floodplains or flood retention areas have not been used in a flood-adapted manner. For example, in August 2002, the intended flooding of the Havelpolders at the confluence of the rivers Elbe and Havel caused widespread fish deaths. The flooding submerged the agriculturally used areas and caused the sensitive crops, i.e. maize, to die off. The ensuing decay processes lowered the oxygen content in the water to such an extent that fish could no longer survive (DKKV, 2003). In June 2013, 430 000 ha of agricultural land was flooded (BMF, 2013), but according to the State Office of Nature Conservation and Landscape Management in the Free State of Saxony no fish deaths were noticed (LFULG, 2013).

In the framework of river monitoring programmes, sediment load and water quality are frequently measured. Measurements are augmented during and after (extreme)

flooding, particularly along the middle reach of the river Elbe due to past mining and industrial activities, particularly in the catchment of the river Mulde (Böhme et al., 2005; BfG, 2014). The measurement programme along the middle reach of the river Elbe provides a wealth of data on the quality of sediments, suspended matter and floodwater (BfG, 2014).

In June 2013, an increased sediment load was observed in all main rivers, i.e. Rhine, Danube, Elbe and Weser (BfG, 2014). In the rivers Rhine and Weser, the total load during the flood each amounted to about 20 % of the average annual load. At many gauges on the rivers Elbe and Danube, even higher loads were measured with a maximum of two thirds of the average annual load at the river Danube and even 80 % of the average annual load at a spot on the river Elbe (BfG, 2014).

In a few samples of water and suspended matter, increased concentrations of heavy metals and arsenic were detected, most probably originating from the Ore Mountains (*Erzgebirge*). The loads of heavy metals amounted occasionally to more than 100 % of the annual load in 2012, reaching a maximum at the Magdeburg gauge. The loads were, however, comparable to those during former flood events, i.e. in 2002 and 2006 (BfG, 2014).

In the suspended matter, greatly enhanced amounts of organic pollutants such as hexachlorocyclohexane (HCH) and derivatives of DDT (1,1,1-trichloro-2,2-bis-(*p*-chlorophenyl)ethane), i.e. DDD and DDE, were measured here and there (BfG, 2014). These pesticides had been produced in chemical plants in Bitterfeld-Wolfen until 1973 (DDT) and 1982 (HCH) and process wastes had been dumped nearby in abandoned open pit mines causing severe pollution of soil and groundwater (e.g. Thieken, 2001; Böhme et al., 2005).

In water bodies in Saxony, aggravating pollution was not experienced in June 2013; all the samples inspected were not toxic (LFULG, 2013). In contrast to this, the Bavarian State Office for the Environment (LfU, 2014) reported a high incidence of contamination by heating oil, especially in the area affected by the breach of the embankment at Deggendorf-Fischerdorf (see Fig. 1c). Leaking heating oil from damaged tanks has

repeatedly been observed during flood events in Germany and had already been identified as a major source for environmental damage during the Whitsun Flood of 1999 in Bavaria. As a consequence, one-off mandatory testing was introduced at that time for heating oil storage facilities with storage volumes of 1000–10 000 L located in flood-prone areas; a measure which came into effect on 1 January 2001 (LfU, 2014).

It is noteworthy that oil leakage and contamination not only harms the environment, it also aggravates damage of flooded buildings considerably (see Kreibich et al., 2005; Thieken et al., 2005). In the administrative district of Deggendorf, up to 150 buildings have to be destroyed and newly erected because of oil contamination (Bavarian Parliament, 2014). Since cost-effective and efficient technical fail-safety systems exist that counteract the floating of oil tanks (e.g. Kreibich et al., 2011), homeowners should be better informed about them. Since 2005, the Federal Water Act states that homeowners are obliged to mitigate damage according to their means. Additionally the implementation of fail-safe measures ought to be monitored more consistently by public authorities (LfU, 2014).

## 4   Discussion and recommendations

In this paper, impacts of the flood of June 2013 in Germany are described with regard to the domains that are addressed by the European Floods Directive (2007/60/EC), i.e. human health, economic activities (and assets), cultural heritage, and the environment. The investigation is further guided by the loss indicators proposed by Corbane et al. (2015) and IRDR (2015) for a consistent loss documentation and is complemented by analyses of traffic disruptions and further impacts perceived as important by affected residents and companies.

It is noteworthy that guidelines on disaster losses such as Corbane et al. (2015) and IRDR (2015) are expected to become more important in the future when it comes to the implementation and monitoring of the Sendai Framework for Disaster Risk Reduction 2015–2030 (SFDRR) that was agreed upon in Sendai, Japan, in March 2015 by

the United Nations (UN). In the SFDRR, seven targets to be achieved by 2030 are listed, among others a substantial reduction of (1) (global) disaster mortality, (2) the number of affected people, (3) direct economic losses as well as (4) damage to critical infrastructure and disruption of basic services such as health and educational facilities

(UN-ISDR, 2015). Apart from these targets, four priority areas for action are defined, in which systematically recorded, evaluated, shared and publicly accessible loss data play a vital role to understand and consequently mitigate the impacts of such events (UN-ISDR, 2015).

In case of the flood event in June 2013, data and information on the flood impacts

could be presented for all four domains considered as relevant by the European Floods Directive and the Sendai Framework SFDRR. The range of impacts portrayed (from direct to indirect damage caused by operational and traffic interruptions through to health and environmental effects) does convey an impression of the diversity of the impacts that flood events can have at different scales. The used data sets have,

however, some strengths and weaknesses. It should be noted that the availability of survey data and traffic disruptions is rather an exception than a rule and does require high efforts and resources for data collection and data processing that may not be available for many events. Such data do, however, provide detailed insights into impacts on the property scale, their perception etc. The analysis shows that affected

residents perceive psychological stress, reinstatement works and supply problems more seriously than damage to buildings or household contents. With regard to economic activities, traffic and business disruptions are more widespread than damage to economic assets or infrastructure elements. The costs attached to these impacts are, however, currently not assessed and hence not included in the overall damage figures.

Therefore, more efforts are needed to include such impacts in loss documentation – by indicators or in monetary terms.

On the contrary, rough data and information from governmental reports, media articles etc. are generally available for many events. For the flood of 2013, there is a clear emphasis of the national and regional reports on the cost assessment of

**NHESSD**

doi:10.5194/nhess-2015-324

**The flood of June 2013 in Germany: how much do we know about its impacts?**

A. H. Thieken et al.

damaged assets for an application to the European Union Solidarity Fund and for the creation of a national reconstruction fund. With regard to human losses common loss indicators such as the number of dead, missing, injured, and directly affected (or exposed) people are only entirely reported on the national level. In the reports of the subnational levels to the Federal Ministry of Finance (BMF, 2013), many specifications are lacking. It is obvious that human losses are not in the focus of an application to the European Union Solidarity Fund. This lack of information is, however, contrasted by the importance that this damage type is given in European and international agreements and that affected residents attribute to physical and particularly mental health problems caused by the flooding. A more comprehensive documentation of human losses together with information on their contexts would be helpful to prevent such losses in future. This was partly undertaken by GMLZ (2014). Most of this information is, however, not publicly accessible as is requested by the Sendai Framework (SFDRR).

Damage to economic activities and particularly to economic assets is the domain for which the most information is available, also on a subnational level since this was the focus of the report by BMF (2013). However, the annexes to BMF (2013) as well as Tables 1 and 2 illustrate that the reports of the affected states to the Federal government differ in length, content and comprehensiveness. Some of the differences can be explained by the different relevance the flood had for the respective state, some by previously experienced flooding or a lack of experience with event documentation. In the future, economic indicators should by default be accompanied by information on the number of damaged or destroyed items (physical damage indicators), such as damaged buildings, enterprises, schools and health facilities as proposed by Corbane et al. (2015). On the one hand, such information will reveal further impacts on the affected population (e.g. supply problems, access to health facilities). On the other hand, it will allow a better comparison and evaluation of the quality of the financial loss estimates reported by the states. The reasonability of first estimates could roughly be appraised by multiplying the numbers of damaged or destroyed item by an average loss per item. Reasonable average losses could be derived from the survey data used

in this paper. In addition, such an approach could help to balance different experiences with event documentation between states. It is of course not applicable to complex damages.

Furthermore, economic sectors should be clearly defined and agreed upon so that the loss documentation of different states and for different events can be better compared. Finally, costs for emergency services and disaster response should be explicitly reported since costing and reporting procedures have been well established in civil protection. The loss reports on the 2013 floods suggest that this potential has not been fully exploited. Only some federal states explicitly provided costs of emergency services of the flood of 2013, others included them to infrastructure losses. This practise should be avoided since in the scientific literature these costs are sometimes regarded as indirect costs (e.g. van der Veen et al., 2003).

Further indirect costs due to traffic or business interruption are currently not included in the overall losses, but might be substantial. Our analysis reveals that traffic disruptions were widespread in 2013 and lasted partly for several weeks and even months. The descriptive assessment of the information gathered on traffic disruptions and interferences illustrate the consequences of the flood without going into the further effects this had on travel times, cancellations of trips or the monetisation of these impacts. The obstruction of shipping, road and railway traffic that is portrayed here was not taken into account in the loss specifications of the states that was presented in Sect. 3.2 and therefore complements the description of flood impacts on (economic) activities. Still, more efforts are needed to derive financial losses of such impacts.

The survey among flood-affected companies further reveals that disruption of production processes and other operations is the most frequently reported flood impact. Since methods to estimate the costs attached to this are in their infancies, this domain requires more attention in research. Data collected on the scale of individual companies can help to derive more reliable estimation models.

In June 2013, damage to cultural assets and heritage accounted only for a small share of the overall financial losses. In consistency with the guideline of Corbane

**NHESSD**

doi:10.5194/nhess-2015-324

**The flood of June 2013 in Germany: how much do we know about its impacts?**

A. H. Thieken et al.

Discussion Paper | Discussion Paper | Discussion Paper | Discussion Paper |

et al. (2015), damaged historic, cultural and heritage places should be explicitly listed together with the scale of their importance, i.e. for the regional, national or international heritage.

With regard to environmental impacts, many measurements of sediment loads as well as of water and sediment quality are available. In 2013, a particular monitoring programme was launched at the middle reaches of the river Elbe due to past mining and industrial activities in the catchment of the river Mulde. In order to better evaluate these measurements, indicators should be developed which also assess the consequences of such contaminations. It is striking that environmental impacts were only addressed in the reports of the water authorities, although contamination by leaking oil tanks is a frequently observed and important driver for building damage (DKKV, 2015). In Germany, the number of floating and leaking oil tanks could thus serve as an important indicator for environmental damage. Since cost-effective and efficient technical fail-safety systems exist that counteract the floating of oil tanks (e.g. Kreibich et al., 2011), homeowners should be better informed about them. Their obligation to mitigate loss should be emphasized. Additionally, the implementation of fail-safe measures ought to be monitored more consistently by public authorities (LfU, 2014).

Altogether, it has to be concluded that the information provided in governmental reports from Germany hardly meet the requirements of European (Corbane et al., 2015) or international (IRDR, 2015) guidelines for disaster loss documentation and databases. Tables 1 and 2 illustrate that more efforts are needed to reach comprehensive loss documentations that are also required for reporting on the progress of the implementation of the SFDRR. Present data and information on flood impacts in Germany appear to be fragmentary, incomplete, partly still preliminary and more often than not publicly inaccessible even for an extreme event such as the flood in June 2013. Since floods are the second most damaging natural hazard in Germany and insurance penetration is still low, transparent and systematic reporting procedures of flood impacts and a related database should be developed.

**NHESSD**

doi:10.5194/nhess-2015-324

**The flood of June 2013 in Germany: how much do we know about its impacts?**

A. H. Thieken et al.

As a minimum effort, a template should be created that is not only usable for applications to the European Union Solidarity Fund but also fulfils minimum requirements of Corbane et al. (2015) and the SFDRR. This template should be generated and agreed upon before the next flood happens and should be accompanied by more robust methods and procedures for first loss estimations.

Ideally, such efforts should be embedded in a broader risk management context in order to not only monitor, but to reduce losses in the longer term. Investment decisions on risk reduction should be combined with an integrated risk management and their effects should be monitored and evaluated. Therefore, an information system on flood impacts and costs should ideally include all relevant cost categories including costs for response and prevention (see Meyer et al., 2013; Kreibich et al., 2014). Hazard information should be clearly linked to data on damage and losses, preferably on an event basis with sub-national spatial resolution. Data collection and provision should be established as a continuous task and enforced by national legislation as (potential) data providers are often non-governmental entities, e.g. with regard to infrastructures. To ensure quality, data collection should be based on transparent rules and methodologies. The set-up of such a system can be done stepwise, but data gaps should be closed gradually. Needed research efforts should be systematically identified and funded and good/best practise examples should be studied and maintained. Only then, event impacts and the effectiveness of the risk reduction measures in place can be reliably evaluated.

## 5 Conclusions

At present, a lack of adequate cost assessment approaches and data on flood impacts limits our knowledge and understanding of appropriate prevention and risk management measures. In comparison to other scientific fields related to the hydrologic system, impact data are still scarce and methods on assessing losses and damage are in their infancies. Therefore, this paper explored what data is available to describe and

**NHESSD**

doi:10.5194/nhess-2015-324

**The flood of June 2013 in Germany: how much do we know about its impacts?**

A. H. Thieken et al.

Discussion Paper | Discussion Paper | Discussion Paper | Discussion Paper |

quantify the impacts of the flood in June 2013, which was the most widespread flooding Germany witnessed over at least the past 60 years (Merz et al., 2014).

The analysis shows that information about impacts in all four domains that are addressed by the European Floods Directive, i.e. human health, economic activities (and assets), cultural heritage, and the environment, is available, but considerably differs in detailedness, completeness and accuracy. The analysis further reveals that drawing up a balance sheet for the impacts of the event in June 2013 has not yet been completed in its entirety. It is further evident that the information currently available does not meet the standards for loss documentation that were proposed by Corbane et al. (2015) for member states of the European Union. Therefore, the establishment of national (and regional) disaster-related accounting systems should be further encouraged. In such an information system, all relevant cost categories including expenditures for risk reduction and response should be included. Furthermore, the system should enable a linkage of flood event indicators with (various) impact indicators in order to evaluate the success of (flood) risk management strategies and measures on the long run. Such an evaluation is required, for example, in progress reports on the Sendai Framework for Disaster Risk Reduction 2015–2030 (SFDRR) that was agreed in Sendai, Japan in March 2015. Only accurate, consistent and comparable databases will allow Germany to substantially and seriously contribute to these internationally agreed targets and commitments.

The range of damages portrayed (from direct to indirect damages from operational and traffic interruptions through to health and environmental effects) conveys an impression of the diversity of the impact that flood events can have. Data collected on the scale of individual properties reveal that business disruption is the most frequently reported damage by affected companies and mental health issues as well as supply problems are perceived more seriously by affected residents than building damage or other forms of financial damage. These damage types receive, however, only little attention in governmental reports on the flood of 2013 as well as in research. However, in the case of evaluating and accepting preventive and protective strategies, these can

**NHESSD**

doi:10.5194/nhess-2015-324

**The flood of June 2013 in Germany: how much do we know about its impacts?**

A. H. Thieken et al.

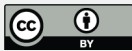

play an important or even decisive role. Therefore, efforts in these domains, starting from data collection to properly describe and understand the phenomena up to effective management strategies in order to reduce these impacts are needed.

*Acknowledgements.* The research presented in this paper was partly conducted by the Forensic Disaster Analysis (FDA) Task Force of the Center for Disaster Management and Risk Reduction Technology (CEDIM) in Potsdam and Karlsruhe and partly in the framework of the project "Hochwasser 2013" funded by the German Ministry of Education and Research (BMBF; funding contracts 13N13016 and 13N13017). Data provision by all ministries and organisations mentioned in the paper is gratefully acknowledged. We further acknowledge the support of Deutsche Forschungsgemeinschaft (German Research Foundation) and Open Access Publication Fund of the Potsdam University.

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

**Table 1.** Overview of human loss indicators as recommended by Corbane et al. (2015) or IRDR (2015) accessible for the flood in June 2013 per federal state (data sources: BMF, 2013 including annexes; GMLZ, 2014 without annexes; ND: no data reported).

| Federal State | Number of people | | | | | Number of helpers |
|---|---|---|---|---|---|---|
| | Died | Missed | Injured | Affected | Evacuated | |
| Baden-Wuerttemberg | human damage is mentioned, but not reported in numbers | | | ND | at least 200 | 18 394 |
| Bavaria | 2 | 0 | 9 | 80 000 | 13 600 | 40 000 |
| Brandenburg | ND | ND | ND | 25 000 | 3500 | ND |
| Hamburg | ND | ND | ND | ND | ND | ND |
| Hesse | ND | ND | ND | ND | ND | ND |
| Mecklenburg- Western Pomerania | ND | ND | ND | ND | ND | ND |
| Lower-Saxony | ND | ND | ND | ND | ND | ND |
| Rhineland-Palatinate | ND | ND | ND | ND | ND | ND |
| Saxony | 1 | ND | 21 | ND | 33 700 (8270 to 16 000 per day) | 76 161 |
| Saxony-Anhalt | 5 | ND | ND | ND | 88 000 | > 120 000 |
| Schleswig-Holstein | ND | ND | ND | ND | ND | 660 |
| Thuringia | ND | ND | ND | ND | ND | ND |
| Overall | 14 | ND | 128 | 600 000 | 80 630 | 871 000 regional and 217 000 federal helpers (in person days) |

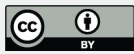

**NHESSD**

doi:10.5194/nhess-2015-324

The flood of
June 2013 in
Germany: how much
do we know about its
impacts?

A. H. Thieken et al.

**Table 2.** Damage and loss indicators as recommended by Corbane et al. (2015) and IRDR (2015) available for the flood of June 2013 in Germany per federal state (data sources: BMF, 2013; Saxon State Chancellery, 2013; Brandenburg, K. Kijewski-Borgel, personal communication, May 2014; Federal Parliament, 2015; ND: no data reported).

| Federal State | Number of damaged or destroyed | | | Overall financial loss | |
|---|---|---|---|---|---|
| | Houses | educational centres (e.g. schools, kindergartens) | health facilities (e.g. hospitals) | reported in BMF (2013) [EUR million] | funds claimed by 30 Jun 2015 (Federal parliament, 2015) [EUR million] |
| Baden-Wuerttemberg | 3697 | 129 | ND | 74 | 59 |
| Bavaria | 13 000 | ND | ND | 1308 | 760 |
| Brandenburg | 1100 | ND | ND | 92 | 81 |
| Hamburg | 0 | 0 | 0 | 1 | 0 |
| Hesse | ND | ND | ND | 21 | 6 |
| Mecklenburg-Western Pomerania | 0 | ND | ND | 8 | 6 |
| Lower-Saxony | ND | ND | ND | 64 | 41 |
| Rhineland-Palatinate | 0 | ND | ND | 4 | 6 |
| Saxony | 13 000 | widespread disruption, no numbers reported | no disruptions in hospitals | 1923 | 1171 |
| Saxony-Anhalt | ND | ND | ND | 2699 | 1496 |
| Schleswig-Holstein | ND | ND | ND | 25 | 14 |
| Thuringia | ND | ND | 1 hospital (power failure) | 452 | 187 |
| Federal Government | – not applicable – | | | 1484 | 114 |
| Emergency response | | | | | 71 |
| Insured loss | | | | | 1650 |
| Total | > 32 000 | disruption mentioned, no numbers reported | ND | 8154 | 5664 |

**NHESSD**

doi:10.5194/nhess-2015-324

The flood of June 2013 in Germany: how much do we know about its impacts?

A. H. Thieken et al.

**Table 3.** Duration (in days) of exceedance of the highest navigable water level at selected gauges on Federal waterways (data source: BfG, 2014, p. 152).

| River (Federal waterway) | Gauge | Duration with water levels above the highest navigable water level [days] |
|---|---|---|
| Moselle | Trier | 6 |
| Rhine | Maxau | 6 |
| | Kaub | 5 |
| Neckar | Heidelberg | 5 |
| Main | Wuerzburg | 9 |
| | Frankfurt | 1–2 |
| Weser | Porta | 10 |
| Danube | Hofkirchen | 15 |
| Elbe | Dresden | 14 |
| | Magdeburg | 12 |
| | Hohnstorf | 11 |

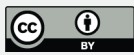

**NHESSD**

doi:10.5194/nhess-2015-324

**The flood of June 2013 in Germany: how much do we know about its impacts?**

A. H. Thieken et al.

**Table 4.** Financial losses of companies affected by the flood in June 2013.

| Loss type | Number of surveyed companies ($n$) | Mean financial loss [EUR] | Median of the financial losses [EUR] |
|---|---|---|---|
| Business disruption | 358 | 137 287 | 15 000 |
| Damaged equipment | 327 | 287 126 | 20 000 |
| Damaged buildings | 310 | 524 292 | 80 000 |
| Damage to goods, products, and stocks | 238 | 46 897 | 8000 |
| Damaged vehicles | 26 | 26 765 | 16 500 |

Discussion Paper | Discussion Paper | Discussion Paper | Discussion Paper |

**NHESSD**

doi:10.5194/nhess-2015-324

The flood of June 2013 in Germany: how much do we know about its impacts?

A. H. Thieken et al.

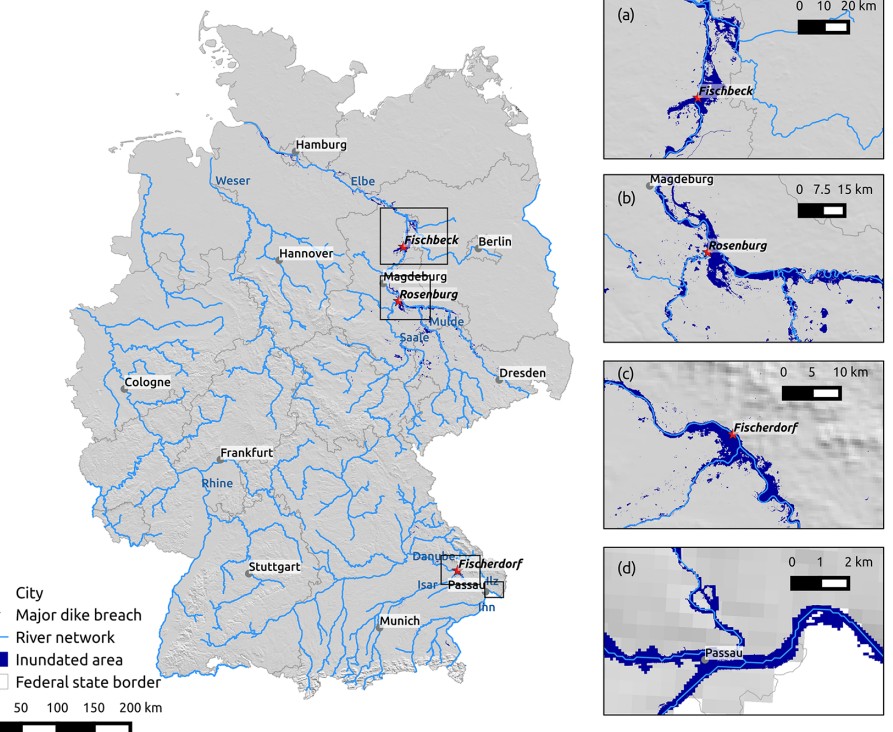

**Figure 1.** Areas inundated in June 2013 and major dike breach locations; details for **(a)** Fischbeck, **(b)** confluence of the rivers Saale and Elbe, **(c)** Fischerdorf at the confluence of the rivers Isar and Danube as well as **(d)** the city of Passau (source: Schröter, 2015, based on satellite images of TerraSAR-X and MODIS).

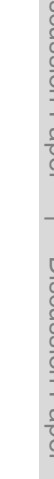

**NHESSD**

doi:10.5194/nhess-2015-324

**The flood of June 2013 in Germany: how much do we know about its impacts?**

A. H. Thieken et al.

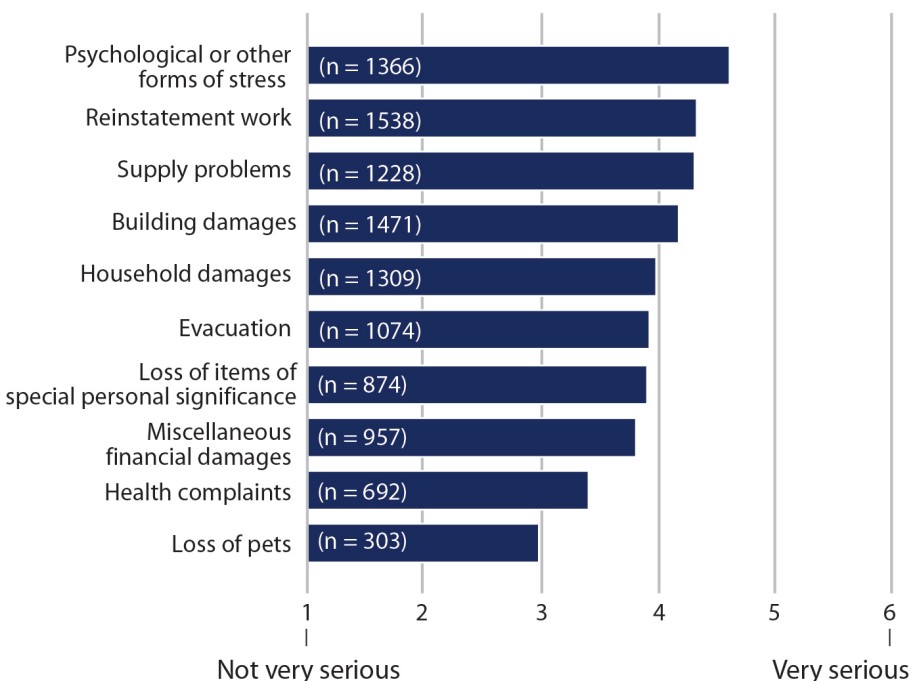

**Figure 2.** Average perception of flood damage witnessed by flood-affected private households and assessed on a scale of 1 (= damage was not very serious) to 6 (= damage was very serious).

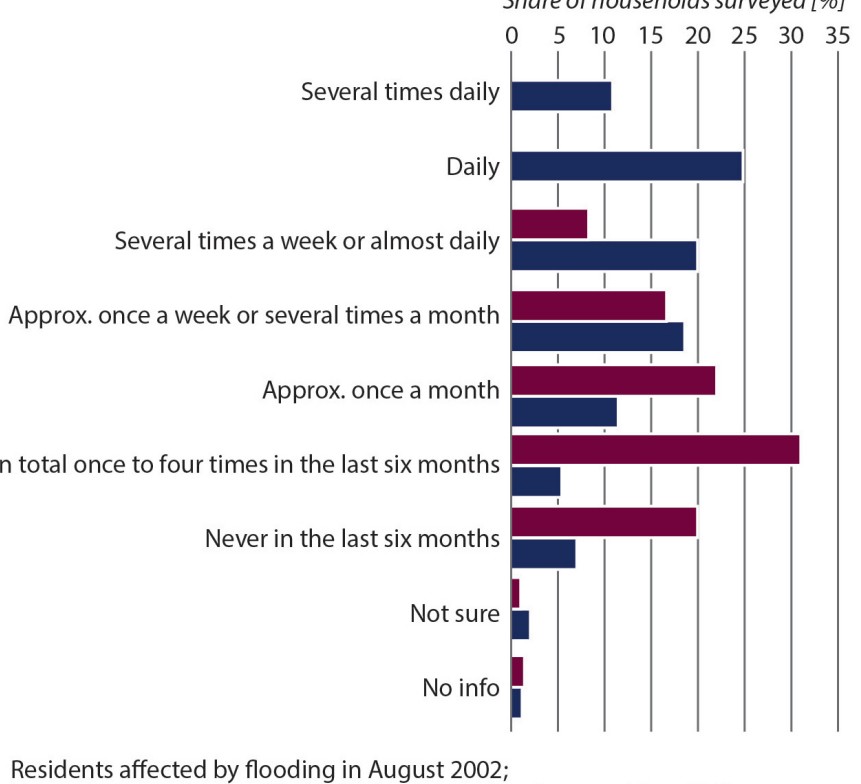

**Figure 3.** Frequency of the flood memories of affected private households in the six months preceding the survey (information is given in percentages of respondents; the first two categories of answers – (several times) daily – were not provided to the respondents in autumn 2012).

NHESSD

doi:10.5194/nhess-2015-324

The flood of June 2013 in Germany: how much do we know about its impacts?

A. H. Thieken et al.

Discussion Paper | Discussion Paper | Discussion Paper | Discussion Paper |

**NHESSD**

doi:10.5194/nhess-2015-324

The flood of June 2013 in Germany: how much do we know about its impacts?

A. H. Thieken et al.

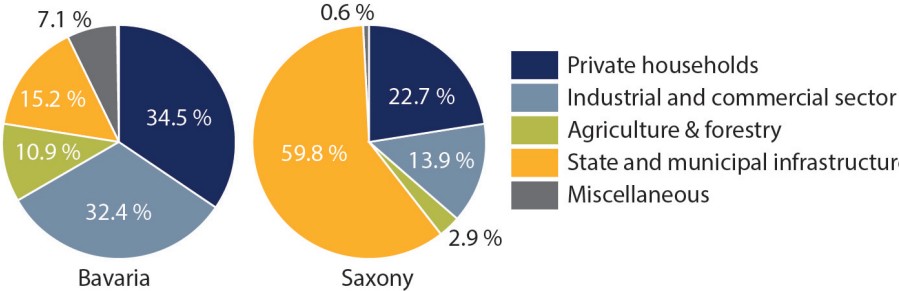

**Figure 4.** Distribution of the overall direct losses of the flood event in June 2013 according to loss-incurring sectors in the federal states of Bavaria (EUR 1.3 billion) and Saxony (EUR 1.9 billion) according to the Federal Ministry of Finance (BMF, 2013).



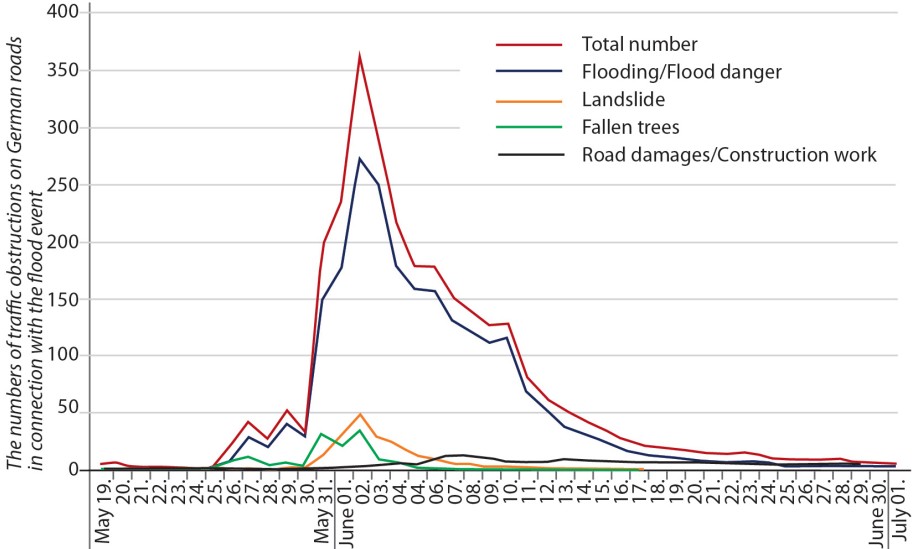

**Figure 5.** Chronological sequence of the number of traffic obstructions on German roads related to the flood event in the period from 19 May to 1 July 2013, subdivided into causes and as a total number.

Discussion Paper | Discussion Paper | Discussion Paper | Discussion Paper |

**NHESSD**

doi:10.5194/nhess-2015-324

**The flood of June 2013 in Germany: how much do we know about its impacts?**

A. H. Thieken et al.

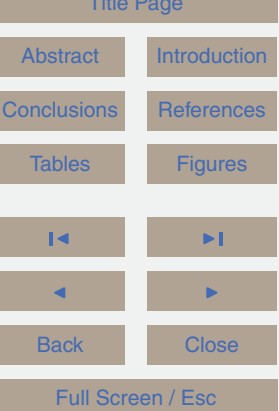

**NHESSD**

doi:10.5194/nhess-2015-324

**The flood of June 2013 in Germany: how much do we know about its impacts?**

A. H. Thieken et al.

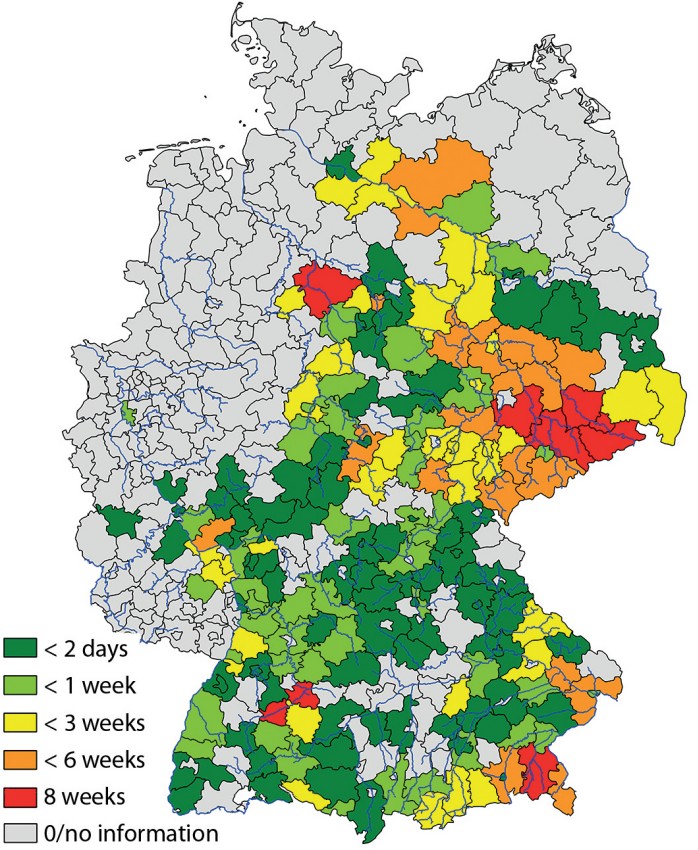

Overall duration of the obstructions in road traffic induced by the flood event shown in terms of administrative districts

- ■ < 2 days
- ■ < 1 week
- ■ < 3 weeks
- ■ < 6 weeks
- ■ 8 weeks
- ■ 0/no information

**Figure 6.** Overall duration of the obstructions in road traffic induced by the flood event shown in terms of administrative districts.

Discussion Paper | Discussion Paper | Discussion Paper | Discussion Paper

**NHESSD**

doi:10.5194/nhess-2015-324

**The flood of June 2013 in Germany: how much do we know about its impacts?**

A. H. Thieken et al.

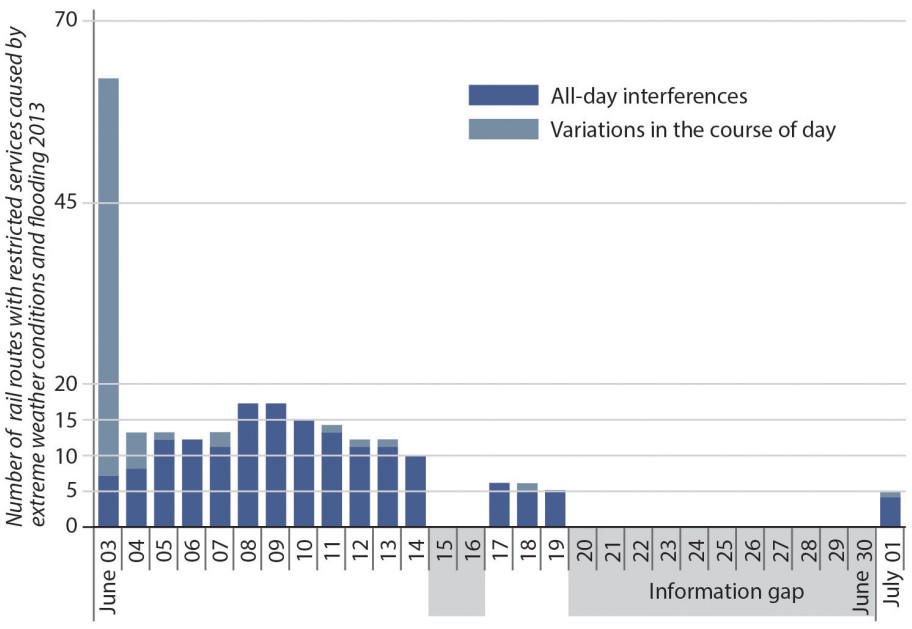

**Figure 7.** Number of train routes with disruptions or interferences caused by extreme weather conditions (low-speed routes, platform or route closures; Information source: German Railways Corporation's internal survey maps detailing interferences caused by extreme weather, in part updated several times a day).

## NHESSD

doi:10.5194/nhess-2015-324

**The flood of June 2013 in Germany: how much do we know about its impacts?**

A. H. Thieken et al.

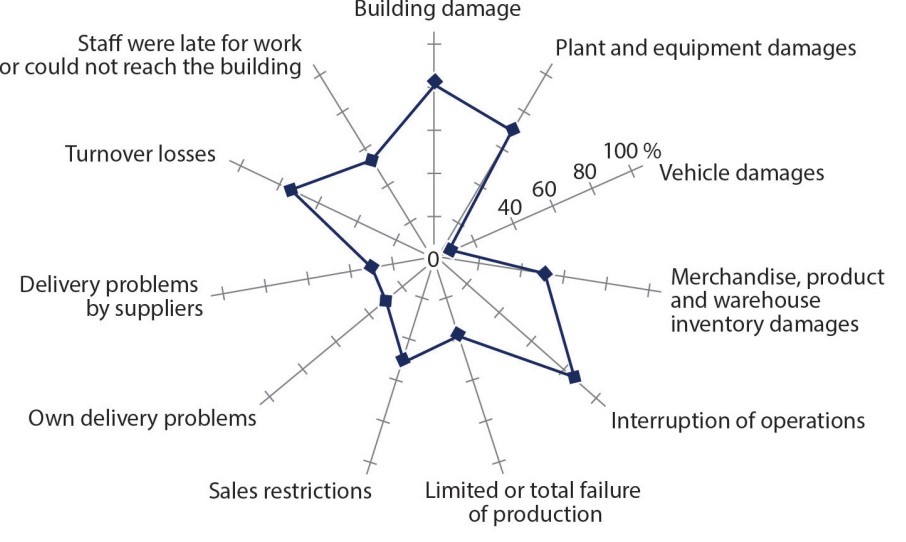

**Figure 8.** Share of surveyed companies that reported on the different flood impacts.