# Peer review of "The flood of June 2013 in Germany: how much do we know about its impacts?"

_Natural Hazards and Earth System Sciences, 2015_

## Referee Comment (RC1) · F. Wenzel (Referee) · 5 Feb 2016

This paper is an excellent study of losses in Germany that resulted from the 2013 flood in Central Europe. In addition to the collection of a huge amounts of information and in addition to surveys executed with people and businesses affected the paper provides a systematic approach to loss and demonstrates its value. This systematic approach includes (a) various risk categories (direct loss, indirect loss, health, transportation) and follows (b) a scheme that has been developed by the European Joint Research Center (JRC). This re-port is an excellent example, one of the few existing ones, where this type of systematic approach is followed and not just data collected that happen to be available. The authors convincingly demonstrate the value of evaluation with this methodology. They also make the point that this type of systematic approach should be developed by scientists, has been developed by JRC and endorsed by a larger

scientific community through IRDR (Integrated Research on Dis-aster Risk, Peking), however, the collection of this type of data should be organized by public institutions.

The paper is very well written, understandable to a broad audience and still of high scientific value. It should be published as it is.

---

## Referee Comment (RC2) · F. Farinosi (Referee) · 10 Mar 2016

F. Farinosi (Referee)

fabio.farinosi@gmail.com

General Comments:

This article is a significant contribution to the long-standing and far from being solved issue of natural disasters' losses quantification. The study took into consideration an extensive flood event occurred in Germany in the recent past. The authors collected the main information about direct human and economic losses reported by national and regional institutions. The study also represented the magnitude of indirect and intangible impacts by presenting proxy data (as for instance the interruptions on the transportation routes) and conducting surveys (mainly for health and psychological impacts). Main objective of the study was to compare what actually the German institutions reported after the 2013 flood with the requirements of the most recent European and interna-

tional guidelines for loss data collection aligned with the objectives of the 2015 Sendai framework on Disaster Risk Reduction. The article concluded that this case study highlighted how the procedures used by the institutions for natural disasters' losses data collection were far from being adequate to the elevated standards required by the most recent literature. The scientific significance of this paper is mainly represented by the effort made in organizing a series of fragmented and heterogeneous information in a more synthetic and comprehensive document that could serve as term of reference for future analysis of institutional development in disaster losses reporting. Doing this, the article fits the scopes of the Natural Hazard and Earth System Sciences Journal. The amount of information collected is massive and denoted an extensive work, but the different sources and the sometimes-incoherent data made its presentation slightly unclear and hard to follow. The authors made no attempt of conducting an assessment exercise aimed at creating new and coherent estimates able to validate or prove the inconsistencies of the official reports. The surveys conducted were aimed at generating data not reported in the official reports. In conclusion, the good contribution that this article brings to the natural hazard scientific community could be further improved with some revisions.

Specific Comments:

- The international reader, with a limited knowledge about the subnational political organization of the country object of the study, would benefit from a better geographic organization (maps) of the majority of the data presented in the article. Figure 1 showed only part of the geo-referenced information exposed in the rest of the article. As an example, in sections 3.2.1, the Authors pointed out that 12 out of 16 federal states were affected by the flooding, and the state of emergency was declared only in 8. This information should probably be reported earlier in the manuscript (Introduction) and the federal states presented in a map.

- In figure 1 the inundated areas should be better highlighted in order to give to the reader a better idea of the flood extent in the each of the federal states.

- A map overlaying the population distribution and the flooded areas in the federal states would help to better represent the exposure to the specific flood event, and the magnitude of the exposed population, in comparison with the "affected" population of the official reports.

- One of the main purposes of the paper was to verify if the information collected by the federal governments covered the main indicators listed in the cited guidelines (De Groeve et al. (2014); Corbane et al. (2015), and IRDR (2015)). A more organized list of the main indicators could help the reader to understand the main requirements and the differences between the federal governments in loss reporting. This was only partially done in Tables 1 and 2.

- In section 2.3.1, the Authors specified that the survey was conducted among households in the flooded areas. Information (map or statistics) about the sample spatial distribution would be helpful to understand the uncertainty that sample size and characteristics brought to the final results.

- The regional stratification of the interviewed subject would help to better present also the information in figure 2. Are respondents affected differently in different areas?

- In section 3.1.1, the description of the table 1 presents numbers that are inconsistent with the table itself. For example, the section reports one human loss in Bavaria, while the table reports 2 losses, etc...

- In section 3.2.1, page 22, the authors described figure 4, comparing the sectoral losses of two of the most affected federal states: in Saxony, state owned infrastructures seem to be more affected than private households, while in Bavaria is the opposite. Is this due to reporting problems (i.e. damages in private households were better documented in Bavaria)?

- Table 3 and Figure 7 (waterways and railways interruptions) would be better if presented on a map, like figure 6.

[Figure]

- Section 3.4 and Figure 8. The results of this survey could be different if the data were stratified by economic sector/industry or, in general, company characteristics (as specified in section 3.1.2).

- Section 3.6, page 32, it would be of use to present the information about pollution spreading on a map.

- A general revision of language and typos (as for instance page 4 "anew") should be also considered.

---

## Author Comment (AC1) · 25 Apr 2016

Hereby, we would like to respond point-by-point to the comments of referee 1.

Comments of referee 1: This paper is an excellent study of losses in Germany that resulted from the 2013 flood in Central Europe. In addition to the collection of a huge amount of information and in addition to surveys executed with people and businesses affected the paper provides a systematic approach to loss and demonstrates its value. This systematic approach includes (a) various risk categories (direct loss, indirect loss, health, transportation) and follows (b) a scheme that has been developed by the European Joint Research Center (JRC). This report is an excellent example, one of the few existing ones, where this type of systematic approach is followed and not just data collected that happen to be available. The authors convincingly demonstrate the value

of evaluation with this methodology. They also make the point that this type of systematic approach should be developed by scientists, has been developed by JRC and endorsed by a larger scientific community through IRDR (Integrated Research on Disaster Risk, Peking), however, the collection of this type of data should be organized by public institutions. The paper is very well written, understandable to a broad audience and still of high scientific value. It should be published as it is.

Our response: It was a pleasure for us to read your comment. Thank you very much for this positive evaluation of our paper and the appreciation of our work.

---

## Author Comment (AC2) · 25 Apr 2016

Hereby, we would like to respond point-by-point to the comments of referee 2.

Comments of referee 2:

General Comments: This article is a significant contribution to the long-standing and far from being solved issue of natural disasters' losses quantification. The study took into consideration an extensive flood event occurred in Germany in the recent past. The authors collected the main information about direct human and economic losses reported by national and regional institutions. The study also represented the magnitude of indirect and intangible impacts by presenting proxy data (as for instance the interruptions on the transportation routes) and conducting surveys (mainly for health and psychological impacts). Main objective of the study was to compare what actually

the German institutions reported after the 2013 flood with the requirements of the most recent European and international guidelines for loss data collection aligned with the objectives of the 2015 Sendai framework on Disaster Risk Reduction. The article concluded that this case study highlighted how the procedures used by the institutions for natural disasters' losses data collection were far from being adequate to the elevated standards required by the most recent literature. The scientific significance of this paper is mainly represented by the effort made in organizing a series of fragmented and heterogeneous information in a more synthetic and comprehensive document that could serve as term of reference for future analysis of institutional development in disaster losses reporting. Doing this, the article fits the scopes of the Natural Hazard and Earth System Sciences Journal.

The amount of information collected is massive and denoted an extensive work, but the different sources and the sometimes-incoherent data made its presentation slightly unclear and hard to follow. The authors made no attempt of conducting an assessment exercise aimed at creating new and coherent estimates able to validate or prove the inconsistencies of the official reports. The surveys conducted were aimed at generating data not reported in the official reports. In conclusion, the good contribution that this article brings to the natural hazard scientific community could be further improved with some revisions.

[Our response]: Thank you very much for this positive evaluation of our paper. In addition, your comments helped us to further improve the manuscript.

It's true that we didn't undertake validations of existing damage reports. We rather focussed on complementing existing reports with information about so far missing damage categories, like transportation interruption, business interruption or health effects. Besides presenting the results for the 2013-flood we also provide an idea about how these often neglected categories could be assessed in future investigations. Thus, we collect, process, analyse and present different, partly unique data that would be inaccessible without this contribution. As such, a better, more complete picture about the

flood impacts arises.

Specific Comments:

[Comment 1]: The international reader, with a limited knowledge about the subnational political organization of the country object of the study, would benefit from a better geographic organization (maps) of the majority of the data presented in the article. Figure 1 showed only part of the geo-referenced information exposed in the rest of the article. As an example, in sections 3.2.1, the Authors pointed out that 12 out of 16 federal states were affected by the flooding, and the state of emergency was declared only in 8. This information should probably be reported earlier in the manuscript (Introduction) and the federal states presented in a map.

[Our response]: In the revised paper, a new map will be added that illustrates the names of the federal states, their degree of being affected by the flood of 2013 (none, slightly, seriously = state of emergency) along with the number of surveyed households and companies (according to your comment 5) so that the information in the Tables 1 and 2 becomes better accessible for an international audience and the spatial spread of the surveyed data becomes clearer (according to your comment 5).

The information about the number of affected states etc. will be shifted to the introduction as suggested.

[Comment 2]: In figure 1 the inundated areas should be better highlighted in order to give to the reader a better idea of the flood extent in the each of the federal states.

[Our response]: In order to better visualise the spatial extent of the flood in each of the federal states, the rivers reaches where the discharge of June 2013 exceeded the 10-year flood discharge and the 100-year flood discharge will be highlighted in the revised figure. Unfortunately, it is neither doable nor meaningful to better highlight the inundated areas. The mapped inundated areas were derived from satellite images that were taken during the flood. However, they mainly contain inundated areas along the

big rivers (such as the rivers Elbe and Danube). Smaller inundated areas, especially at the beginning of the event are not captured in the images. Therefore, the mapped inundated areas are far from being complete. In our opinion, they only give an indication of the hot-spots of the flood event.

The figure caption will be adapted accordingly; in its current form it might raise too high expectations.

[Comment 3]: A map overlaying the population distribution and the flooded areas in the federal states would help to better represent the exposure to the specific flood event, and the magnitude of the exposed population, in comparison with the "affected" population of the official reports.

[Our response]: In principle, a map containing the disaggregated population density is available for the whole of Germany (see THIEKEN, A.H., M. MÜLLER, L. KLEIST, I. SEIFERT, D. BORST, U. WERNER (2006): Regionalisation of asset values for risk analyses. – Nat. Hazards Earth Syst. Sci. 6(2): 167-178, http://www.nat-hazards-earth-syst-sci.net/6/167/2006/). An intersection of the mapped inundated areas as shown in Fig. 1 with the map published by Thieken et al. (2006) could hence be performed easily. However, due to the above-mentioned restrictions and gaps of the mapped inundated areas (Fig. 1), we don't think that such an exercise would deliver a reliable number of exposed people. Therefore, we refrain from doing this.

[Comment 4]: One of the main purposes of the paper was to verify if the information collected by the federal governments covered the main indicators listed in the cited guidelines (De Groeve et al. (2014); Corbane et al. (2015), and IRDR (2015)). A more organized list of the main indicators could help the reader to understand the main requirements and the differences between the federal governments in loss reporting. This was only partially done in Tables 1 and 2.

[Our response]: In the sections 3.5 and 3.6 the information items recommended by Corbane et al. (2015) and IRDR (2015) are given in the text. However, we will introduce

a new table that summarizes the main data fields requested by Corbane et al. (2015) and IRDR (2015) and the contributions made for the 2013-flood.

[Comment 5]: In section 2.3.1, the Authors specified that the survey was conducted among households in the flooded areas. Information (map or statistics) about the sample spatial distribution would be helpful to understand the uncertainty that sample size and characteristics brought to the final results.

[Our response]: See comment 1. We will display the spatial distribution of the sample on a new map.

[Comment 6]: The regional stratification of the interviewed subject would help to better present also the information in figure 2. Are respondents affected differently in different areas?

[Our response]: Intentionally, we provided an overview of the full data set since the main intention was to illustrate the range and the ranking of the flood impacts as perceived by the affected people. While revising the paper we will check whether substantial regional differences will occur. If so, this information will be added to the paper.

[Comment 7]: In section 3.1.1, the description of the table 1 presents numbers that are inconsistent with the table itself. For example, the section reports one human loss in Bavaria, while the table reports 2 losses, etc.

[Our response]: We apologize for this mistake. In fact, the data in the table are correct. In the text Bavaria and Saxony were mixed up. Of course, this will be corrected in the revised version.

[Comment 8]: In section 3.2.1, page 22, the authors described figure 4, comparing the sectoral losses of two of the most affected federal states: in Saxony, state owned infrastructures seem to be more affected than private households, while in Bavaria is the opposite. Is this due to reporting problems (i.e. damages in private households were better documented in Bavaria)?

[Our response]: Actually, the reason for these differences is unclear and needs further investigations that are beyond the scope of this paper. Since details about the damaged type of infrastructure are not available for both states, a comparison of the documented damage is almost impossible. An additional hurdle is the lacking documentation of the reporting methodologies. Since the damage documentation and reporting was done by each federal state without a clear guidance from the federal level, a reporting bias cannot be excluded. The different flood insurance penetration in Saxony (>40%) and Bavaria (around 20%) might also play a role, since it is not 100% clear, whether or to what extent insured losses are included in the damage documentations of the states. Doubtlessness about the final share of damage to infrastructures can only be achieved when the final losses and damage are recorded, which cannot be expected by next year.

[Comment 9]: Table 3 and Figure 7 (waterways and railways interruptions) would be better if presented on a map, like figure 6.

[Our response]: We will try to integrate the data presented in Table 3 in Fig. 1. The detailedness of the provided information, however, does not allow us to present the data of Figure 7 as map.

[Comment 10]: Section 3.4 and Figure 8. The results of this survey could be different if the data were stratified by economic sector/industry or, in general, company characteristics (as specified in section 3.1.2).

[Our response]: We agree that different economic sectors are differently affected by a flood event. We will add this analysis for different sectors (i.e. manufacturing, commerce, finance, services and agriculture). For example, own delivery problems and delivery problem by suppliers are the most frequently reported by the manufacturing sector, while sales restrictions are the most frequently reported by the commercial sector. Since damage to vehicles turned out to be negligible for all sectors, this category will be removed (to also maintain the readability of the revised figure).

[Figure]

[Comment 11]: Section 3.6, page 32, it would be of use to present the information about pollution spreading on a map.

[Our response]: Unfortunately, the detailedness of the available data does not allow us to draw a map.

[Comment 12]: A general revision of language and typos (as for instance page 4 "anew") should be also considered.

[Our response]: The manuscript was carefully read again and corrected where necessary. On page 4, "anew" was used in the meaning of "again", not as "a new". The use is correct in this context.